# Linear interaction between replication and transcription shapes DNA break dynamics at recurrent DNA break Clusters

Lorenzo Corazzi [1,2,7], Vivien S. Ionasz [1,2,7], Sergej Andrejev[1], Li-Chin Wang[1], Athanasios Vouzas[3,4], Marco Giaisi[1], Giulia Di Muzio[1,2,5], Boyu Ding [1,2,6], Anna J. M. Marx[1,2,5], Jonas Henkenjohann [1,2,5], Michael M. Allers [1,6], David M. Gilbert [4] & Pei-Chi Wei [1,2,5] ✉

Recurrent DNA break clusters (RDCs) are replication-transcription collision hotspots; many are unique to neural progenitor cells. Through high-resolution replication sequencing and a capture-ligation assay in mouse neural progenitor cells experiencing replication stress, we unravel the replication features dictating RDC location and orientation. Most RDCs occur at the replication forks traversing timing transition regions (TTRs), where sparse replication origins connect unidirectional forks. Leftward-moving forks generate telomere-connected DNA double-strand breaks (DSBs), while rightward-moving forks lead to centromere-connected DSBs. Strand-specific mapping for DNA-bound RNA reveals co-transcriptional dual-strand DNA:RNA hybrids present at a higher density in RDC than in other actively transcribed long genes. In addition, mapping RNA polymerase activity uncovers that head-to-head interactions between replication and transcription machinery result in 60% DSB contribution to the head-on compared to 40% for co-directional. Taken together we reveal TTR as a fragile class and show how the linear interaction between transcription and replication impacts genome stability.

DNA double-strand breaks (DSBs) emerge at stalled forks to reconfigure and restart DNA replication[1]. During the fork remodeling process, the structure-specific DNA nucleases from the XPF family proteins are recruited at stalled forks. MUS81, one of the XPF family proteins, recognizes the Y-shaped DNA junctions and makes cuts[2]. MUS81 is essential for common fragile sites (CFS) expression[3,4] and mitotic DNA synthesis[5], indicating a connection between DNA breaks and late-replicating regions without replication origins. Additionally, it was suggested that replication stress preferentially affects the timing transition region (TTR)[6], characterized by infrequent origin firing and long-traveling replication forks[7–9]. This arrangement, in theory, renders TTR vulnerable to replication stress. Nevertheless, their vulnerability and unidirectional nature in response to replication stress remain unexplored. Furthermore, genome fragility can also be observed in regions adjacent to or within replication initiation zones[10], resulting in early replicating fragile sites (ERFS)[11]. Although mechanisms for ERFS and late-replicating genomic regions are suggested to differ[3,4,11], a direct comparison of DNA break dynamics at these sites has not yet been conducted to support this hypothesis.

Transcription-replication conflicts (TRCs) pose a challenge when transcription and replication occur in the same genomic regions. Long genes are particularly susceptible to TRC in mammalian cells[12–14].

[1]German Cancer Research Center, 69120 Heidelberg, Germany. [2]Faculty of Bioscience, Ruprecht-Karl-University of Heidelberg, 69120 Heidelberg, Germany. [3]Department of Biological Science, Florida State University, Tallahassee, FL 32306, USA. [4]San Diego Biomedical Research Institute, San Diego, CA 92121, USA. [5]Interdisciplinary Center for Neurosciences, Ruprecht-Karl-University of Heidelberg, 69120 Heidelberg, Germany. [6]Faculty of Medicine, Ruprecht-Karl-University of Heidelberg, 69120 Heidelberg, Germany. [7]These authors contributed equally: Lorenzo Corazzi, Vivien S. Ionasz. ✉e-mail: p.wei@dkfz-heidelberg.de

CFSs are affected by TRCs, as they contain long and actively transcribed genes or gene units[12–14]. CFSs are associated with copy number losses[15]. One proposed explanation suggests that CFSs could be situated within "double fork failure" zones[13,15], where two inward-moving replication forks traverse the entire gene, leading to DNA replication termination midway through the gene at the end of the S phase. This "double fork failure" model has been implied as an essential mechanism for copy number loss at CFSs[16]. However, how "double fork failure" exclusively results in deletions is unclear. While it was suggested that the orientation of DNA ends present at the fork might play a role in copy number loss at CFS[15,17], how DSB ends align at a series of colliding forks to promote deletions remains to be determined.

The role of transcription activity during a TRC conflict is primarily studied in genomic regions prone to R-loop accumulation[18]. R-loop persistence has been linked to DNA instability[19], particularly in head-on conflicts where transcription and replication occur in opposing directions[19], while some R-loops arise physiologically and do not promote TRC or DSBs[20]. Experimental models using episomes suggest that head-on collisions enhance R-loop accumulation and DNA breaks[19]. However, it is not clear whether the orientation of TRCs matters in actual chromosomes exhibits similar kinetics, as CFS often have low levels of R-loops[21]. Thus, it remains to be explored whether transient DNA:RNA hybrids associated with transcription are present at TRC sites and if they correlate with fork slowing and DNA breaks.

Recurrent DNA break Clusters (RDCs) were recently described in mouse neural progenitor cells (NPCs) treated with low-dose aphidicolin, a DNA polymerase inhibitor[22,23]. Although the specific genomic positions of RDCs may vary between different cell types, all RDCs consistently localize to actively transcribed genes, making RDC a class of TRC[22,23]. These RDCs were detected through ligation to DSBs induced by exogenous nucleases such as CRISPR/Cas9 and I-Sce-I. Most genomic regions of RDCs replicate at the latter half of the S phase, while some RDC-containing sequences are associated with earlier replication[22,23]. These genes are generally > 100 kb, with a few exceptions, and DNA breaks are distributed across the entire gene body within an RDC-containing gene. Not all actively transcribed genes longer than 100 kb exhibit RDCs, suggesting that transcription activity alone cannot trigger DNA breaks. Whether there are general rules governing the emergence of RDCs requires further investigation.

Our study aimed to investigate the relationship between replication-transcription encounters and the distribution of DNA breaks at the linear scale at the RDC loci. We identified three distinct features regarding RDCs: (1) RDCs primarily align to TTRs containing actively transcribed genes, (2) DNA break density within a subset of RDCs overlap to a late broad constant timing zone, and (3) early replicating RDCs exhibit a strong relationship with R-loop accumulation. We also observed the presence of co-directional DNA:RNA hybrids at the RDCs. These hybrids are associated with a higher density of DSBs at the head-on encounters between replication and transcription than codirectional encounters within RDCs.

## Results

### A comprehensive dataset to align DNA breaks to the linear movement of DNA replication and transcription in mouse neural progenitor cells

RDCs have been mapped across all chromosomes in mouse NPCs, whereas in other cells, only the most prominent RDCs have been successfully mapped[22–24]. To ensure a thorough exploration of the linear interaction underlying the RDC-containing loci, we conducted our subsequent experiments using XRCC4/p53-deficient mouse NPCs. We chose this genotype to recover genome-wide DSBs at RDCs efficiently[22]. In this model, *Tp53* was knocked out to prevent NPC death caused by XRCC4 deficiency[22]. We generated multiple datasets from XRCC4/p53-deficient NPCs derived from ES cells to analyze DNA replication direction, DNA break density, transcription direction, and

DNA:RNA hybrid position on a linear scale within the same genetic background (Supplementary Data 1). The RDC collection (Supplementary Data 2) described in this article is characterized by combining the published[22,23] and newly generated DNA break density datasets. A detailed description of the RDC calling process can be found in the Methods section. We characterized 152 RDCs, 78 of which were described previously (Supplementary Data 2). The additionally characterized RDCs are all in genomic regions containing actively transcribed genes (Supplementary Data 2). Consistently with the findings of previous RDC studies[22,23], genes underlying the additionally identified 74 RDC show an overrepresentation of neuronal functions and encode proteins controlling cell adhesion and synaptic functions (Supplementary Data 3). In this article, we analyzed the relationship between DNA breaks and the linear interaction of genomes under the 152 RDCs.

### Replication direction maps for XRCC4/p53-deficient neural progenitor cells

To examine the directionality of DNA replication within the DNA sequences underlying RDCs, we conducted high-resolution Repli-seq on XRCC4/p53-deficient, ES cell-derived NPCs, with and without aphidicolin treatment (Fig. 1A). In ES cell-derived NPCs, aphidicolin treatment delayed DNA replication timing at several genomic loci (Fig. 1B), aligned with the observation in human lymphoblasts and fibroblasts[6,25]. Additionally, aphidicolin treatment advanced replication timing at particular genomic loci (Fig. 1B) as described previously[6]. The comprehensive investigation of genome-wide replication timing changes before and after aphidicolin treatment has been discussed elsewhere[6,25]. Subsequently, we applied the balanced iterative reducing and clustering using hierarchies (BIRCH) algorithm[26] to determine replication features (Fig. 1C, Supplementary Fig. 1A, and Supplementary Data 4). Consistent with the high-resolution Repli-seq data generated using wild-type mouse NPCs[26], the majority of the mouse NPC genome contains timing transition region (TTR) (38 – 41%; example in Supplementary Fig. 1A), where DNA replication is conducted by long-traveling unidirectional replication forks. The second most prevalent feature is the constant timing region (CTR) (24 – 31 % example in Supplementary Fig. 1A), where multiple replication units rapidly complete DNA replication. In untreated and APH-treated NPC, 16 – 20% of the genome contains initiation zones, and 1–3% contains small termination zones. We determined the fork direction by connecting the initiation zone to the nearest replication termination points assisted by a convolutional neural network (Supplementary Fig. 1B-D and Methods). Replication directions agreed by both technical repeats from the APH-treated NPC were used for downstream analyses.

### Most RDCs are oriented single-ended DSBs at the timing transition region

We hypothesize that RDCs consist of solitary DSB ends at the fork, which means that DSB within RDCs exhibits specific orientations aligned with replication fork directions (Fig. 2A). To determine the orientations of DNA double-strand breaks (DSBs) within RDC, we employed the linear-amplification mediated, high-throughput genome-wide translocation sequencing (LAM-HTGTS)[27]. In this approach, a specific DSB known as the "bait" end is induced by sequence-specific enzymes such as CRISPR/Cas9 ([28] and Supplementary Fig. 2). Translocations between the bait end and other "prey" DSB ends are then identified, and the orientations of these prey DSBs are retained through linear-amplification-mediated PCR (Supplementary Fig. 2). At the stalled replication fork, structure-specific nucleases cut the junction of the Y-shaped DNA, resulting in single-ended DSB (Fig. 2A). These DSBs are either centromeric-oriented (Dcen) or telomeric-oriented (Dtel), depending on the direction of fork progression. DNA break present at the fork as single-ended has been shown before in yeast[29] and mammalian cells[30]. It is important to note that mechanisms

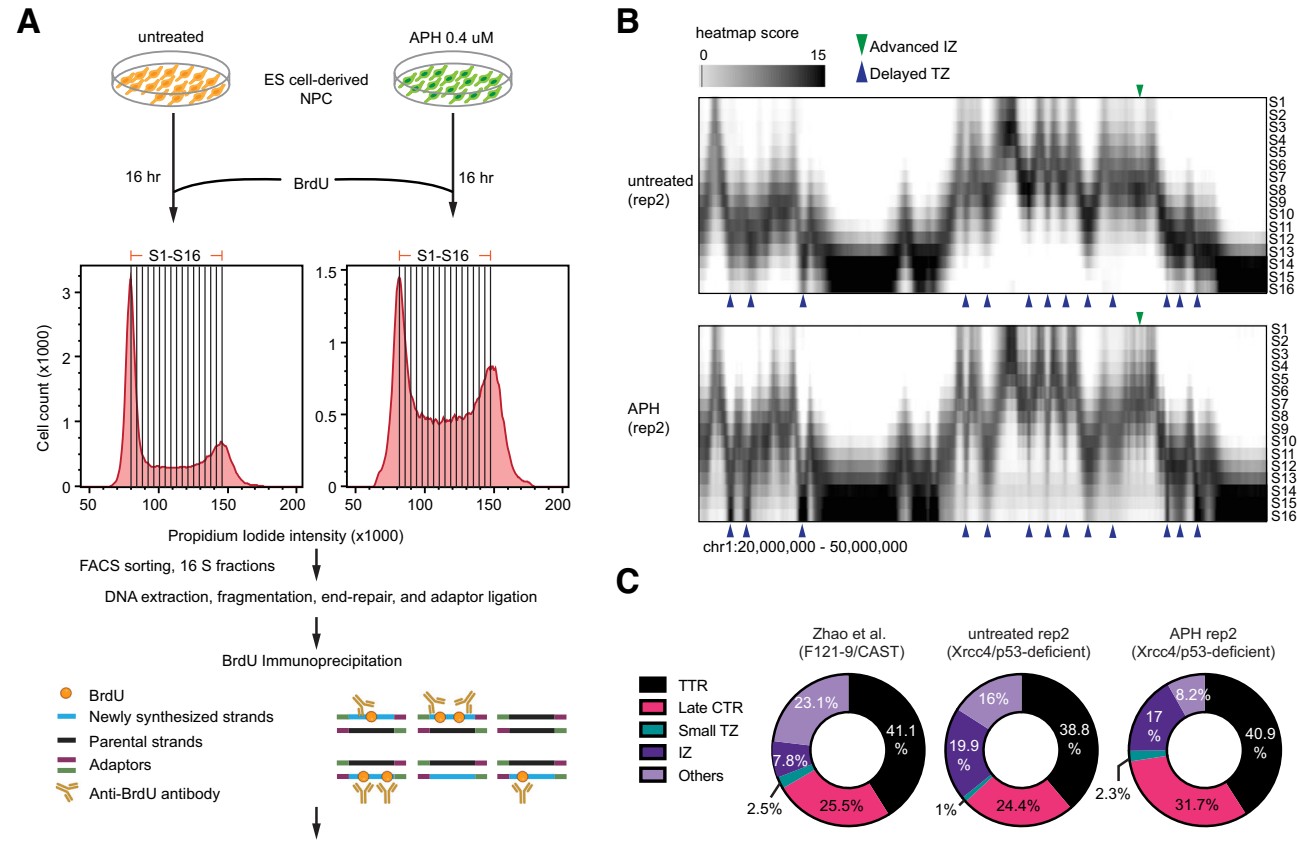

**Fig. 1 | High-resolution Repli-seq unveils replication with precise temporal resolution in mouse ES cell-derived NPCs. A** Overview of the Repli-seq library preparation workflow. The details for cell and library preparation are described in the Methods section. APH: aphidicolin. BrdU: bromo-dUTP. FACS: fluorescence-activated cell sorting. **B** High-resolution Repli-seq normalized and scaled heatmap for chr1:20,000,000–50,000,000. The top panel represents untreated ES cell-derived NPCs, while the bottom represents treated ES cell-derived NPCs. The Y-axis ranges from S1 to S16 fractions. The green arrow highlights the advanced initiation zone in the APH-treated sample compared to the untreated sample, and the blue arrows emphasize the delayed termination zones in the APH-treated sample. IZ: initiation zones. TZ: termination zones. **C** Repli-seq genome profile of untreated ES cell-derived NPCs compared with Zhao et al. 2020 (F121-9/CAST) mouse ES cell-derived NPCs dataset. TTR timing transition region, Late CTR late constant timing region, Small TZ small termination zone, and IZ initiation zone.

other than RDC may be involved in translocations to sites outside of RDCs, such as off-target sites generated experimentally with CRISPR/ Cas or recombining immunoglobulin gene loci, which have been described elsewhere[28,31,32].

Next, we analyzed the orientation of DSBs in aphidicolin-treated NPCs and their alignment with replication fork directions in the mouse genome. Of the 152 RDCs, 87 RDCs consisted of pairs of rightward- and leftward-moving forks converging at a late replicating zone (Fig. 2B–G, Supplementary Figs. 3 and 4). These RDCs are referred to as "inward-moving" RDCs. Among them, 63 overlapped with the TTR. It was proposed that multiple unidirectional forks connected with sparse origins transverse between early replicating regions to the late-replicating regions[8,33,34], such as TTRs at *Npas3* and *Cdk14* span most S-phase fractions (Fig. 2B, C). Thus, long TTR cannot be given a replication timing. Our observations suggest that forks stalling at inward-moving RDCs leads to centromeric DNA break ends (Dcen) at the rightward TTR and telomeric DNA break ends (Dtel) at the leftward TTR. This pattern was consistently observed in loci such as *Npas3* and *Cdk14*, where pairs of Dcen and Dtel peaks colocalized with the TTR at the RDC (Fig. 2C). The peaks were separated by a substantial distance of up to 700 kb and the "twin-peak" pattern was consistent across most inward-moving RDCs associated with TTR. We found exceptions in eight "inward-moving" RDCs (*Tenm3, Mast4, Magi1, Sox6, Pard3, Dock1, Tbc1d5,* and *Adk*) displayed "single-peak" features, five of which aligned to a higher transcription activity at the corresponding transcript isoforms.

We found six "inward-moving" RDC loci (*Sil1, Col4a2, Qk, Zmiz1, Rere,* and *Msi2*) where the genomic sequences underneath were replicated within the earlier S phase fractions (Supplementary Fig. 3). RDCs in these regions lost the "twin-peak" signatures. For instance, Dcen and Dtel largely overlap at *Qk* and *Rere* gene loci. This finding suggests that early replicating genomic regions follow separate TRC mechanisms uncoupled from the fork progressing direction.

We observed 23 "inward-moving" RDCs, such as the *Ctnna2* and *Prkg1* gene loci, displayed large CTRs exceeding 500 kb (Fig. 2D–G). The density of Dcen and Dtel was concentrated at the CTR and showed some degree of overlap, with the interval between the two peaks reduced to 300 kb. Among them, 16 contain TTRs, including RDC at the *Prkg1* locus (Fig. 2D, E). We found that TTR is absent from three RDCs (*Cadm2, Pcdh9,* and *Astn2*), where RDCs are flanked by IZ starting at the median S phase. Similar to RDC at *Prkg1*, the DNA breaks are predominantly enriched at the CTR region in *Cadm2* (Fig. 2F, G) and in other RDCs that do not contain TTR (Supplementary Fig. 3).

In addition to the "inward-moving" RDCs, we found 15 RDCs where the underlying genome was replicated exclusively by rightward- or leftward-moving forks (Fig. 3A, B, and Supplementary Fig. 4). These unidirectional forks coincided with a single rightward or leftward TTR in aphidicolin-treated NPC. We refer to these RDCs as "unidirectional" RDCs. For example, at the *Large* RDC, a right-moving fork traversed the genome beneath the Dcen peak (Fig. 3A). Conversely, a left-moving fork covered the genome beneath the Dtel peak at the *Csmd2* RDC (Fig. 3B). The genome regions underlie the "unidirectional" RDC as well

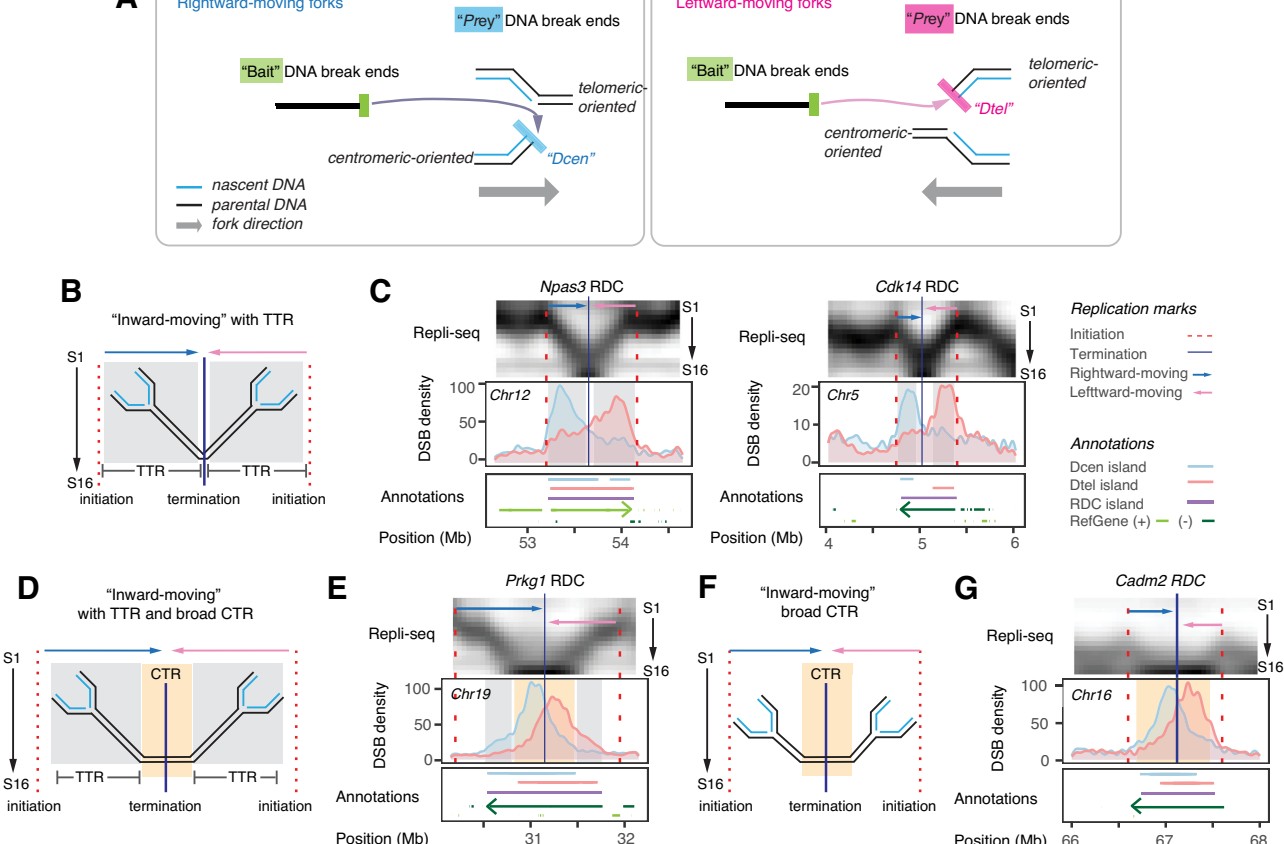

**Fig. 2 | DNA break end orientation of "inward-moving" RDC and their association with the timing transition regions. A** The figure illustrates single-ended DNA breaks at rightward- (left) and leftward-moving forks (right). The light blue DSB end at the rightward-moving fork is linked with centromeres, maintaining its centromeric orientation (Dcen) when joined with the "bait" DSB end. Conversely, pink DSB ends at the leftward-moving forks are linked with telomeric sequences, preserving their telomeric orientation (Dtel) upon joining with the "bait" DSB end. **B** Illustration of inward-moving fork direction at RDC with two timing transition regions (TTRs), shaded in gray. The arrow above each fork annotates the replication direction. Light blue represents rightward movement, and pink indicates leftward movement. Initiation zones correspond to the dashed red line, and the blue line indicates the termination zone. TTRs were shaded in gray rectangular. **C** Multiomics figures provide information regarding genomic loci containing two "inward-moving", TTR-containing RDCs at the *Npas3* and *Cdk14* gene loci. The top panel shows a high-resolution Repli-seq normalized heatmap using aphidicolin (APH)-treated ES cell-derived NPCs, as described in Fig. 1B. Pink and blue arrows indicate the replication directions of the genome beneath RDC. The middle panel displays DNA break density, with the *Y*-axis representing extended interchromosomal Dcen or Dtel density per 25 kilobases kernel. The bottom panel includes annotations indicating Dcen and Dtel island positions, RDC range (purple line), and actively transcribed gene orientation (+: light green, -: dark green). Additional annotations associated with this figure are depicted on the right. **D**, **F** Illustrations of inward-moving fork direction at RDCs with (**D**) or without (**F**) TTR flanking an extended late constant timing region (CTR), which is shaded in yellow. **E**, **G** Multiomics figures describe the genomic loci containing two "inward-moving," CTR-containing RDC at *Prkg1* (**E**) and *Cadm2* (**G**). Figures are shown as described in (**C**).

encompass the TTRs (Supplementary Data 5). Among the 15 "unidirectional" RDCs, 14 exhibited a "single-peak" pattern, where the density of DNA ends aligned with the direction of the respective fork in that region.

Furthermore, we observed RDCs at genomic regions that undergo biphasic replication (Fig. 3C, D, and Supplementary Fig. 4). Biphasic replicating regions are associated with CFS in human cells[26]. TTR also colocalizes with the DSB density within "biphasic" RDCs. The DNA break ends (Dcen and Dtel) peaks primarily aligned with the orientation of the TTR; in some cases, a secondary summit was observed. For instance, the secondary Dcen summit at the *Samd5* and *Auts2* RDCs corresponds with additional "rightward-moving" forks in that region (Fig. 3C, D).

Among the 152 RDCs analyzed, only six RDCs were found in genomic regions replicated by outward-moving forks. These "outward-moving" regions replicate early, similar to ERFS[11]. In these regions, the distributions of Dcen and Dtel at RDCs exhibited overlapping patterns (Supplementary Fig. 4). We speculate that the DNA breaks in early replicating RDCs, including those in large initiation zones, occur through mechanisms different from those at TTR and may be

influenced by other transcription-dependent mechanisms. Lastly, RDCs spanning three or more fork directions are classified as "complex" (Supplementary Data 5, Fig. 3D, and Supplementary Fig. 4). The "complex" class includes RDCs at *Auts2* and *Tenm4*, which display biphasic replication. The remaining 32 RDCs either lack replication features to assist fork direction assignment or are entirely located within late CTR or broad initiation zones and cannot be assigned a specific replication direction. These RDCs are classified as "undetermined" (Supplementary Fig. 4).

By aligning the distribution of Dcen and Dtel with the direction of replication forks, we conclude that most RDCs are DNA breaks occurring at forks passing through TTR, with some exceptions occurring at initiation zones and late CTRs (Fig. 3E).

## The density of the DSB ends corresponds to DNA replication stalling

Next, we investigated whether the density of DSBs correlates positively with the level of replication stress. We analyzed the DSB density at RDCs in NPCs treated with different concentrations of aphidicolin. In the case of the "inward-moving" RDC at the *Prkg1* locus, the DSB

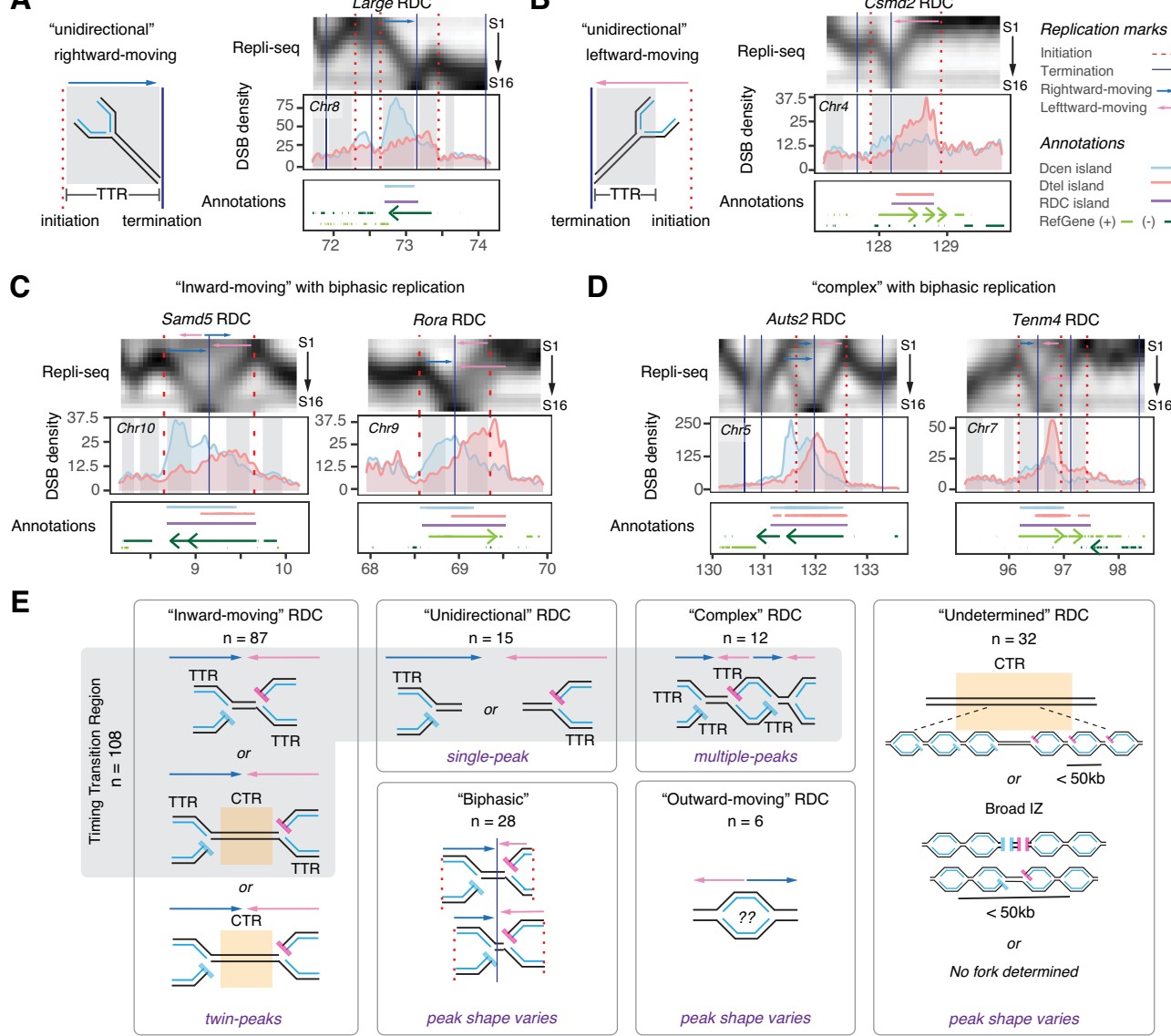

**Fig. 3 | DSB distribution at the corresponding "unidirectional" and biphasic-replicating RDCs. A** Left panel: illustration of a unidirectional rightward-moving fork. The figure is organized as shown in Fig. 2B. Right panel: multiomics data provide information regarding the *Large* locus containing a "unidirectional" RDC in aphidicolin-treated ES cell-derived NPCs. TTR: timing transition region. **B** Left panel: illustration of a unidirectional leftward-moving fork. Right panel: multiomics plot showing the replication, RDC break distribution, and associated annotations at

*Csmd2* locus in ES cell-derived NPCs treated with APH. **C** Multiomics panels show the *Samd5* and *Rora* loci information containing biphasic "inward-moving" RDC. **D** Multiomics panels show the information of the *Auts2* and *Tenm4* gene loci, which contain biphasic "complex" RDC. The multiomics panels are shown as described in Fig. 2C. **E** A panel summarizes RDC types based on DNA replication direction, with the plot indicating the count for each type. Biphasic RDC is a distinctive feature; it does not constitute a separate category. CTR constant timing region.

density increased from six DSBs per ten thousand interchromosomal translocations in untreated NPCs to 19.8 and 24.3 in NPCs treated with 0.3 μM or 0.4 μM aphidicolin (Fig. 4A). This aphidicolin dosage-dependent elevation is consistent with the DSB density at the genome regions containing the "inward-moving" RDCs (Fig. 4B). Similar trends were observed for the "unidirectional" RDC at the *Cep112* locus (Fig. 4C). The DSB count raised from 1.4 per ten thousand interchromosomal translocation in the untreated NPC, to 5.5 and 8.6 in NPCs treated with 0.3 μM or 0.4 μM aphidicolin. The rest of the "unidirectional" RDCs displayed a DSB density increment as in the *Cep112* RDC (Fig. 4D). This trend was also observed in other "unidirectional" RDCs (Fig. 4D), the "complex" RDC located at the *Auts2* locus (Fig. 4E), and other "complex" RDCs (Fig. 4F).

Within the aphidicolin dosage datasets, the DSB density is too low to make a firm conclusion for DNA break changes in the "outward-moving" RDCs. These RDCs were less robust than the other RDC types.

The overall interchromosomal translocation DSB density beneath the genome of "outward-moving" RDCs was much lower, between 1.2 and 4.6 interchromosomal translocations per ten thousand in NPCs treated with 0.4 μM aphidicolin. The DSB count decreased to one per RDC in NPC treated with 0.2 μM aphidicolin. As a result, we did not observe a significant aphidicolin-dependent DSB elevation in the "outward-moving" RDC.

In summary, DNA breaks at the "inward-moving," "unidirectional," and "complex" RDCs increased in an aphidicolin dose-dependent manner, suggesting RDCs represent DNA breaks at the reprogrammed forks.

### RDC displays differential position accordance to DNA:RNA hybrid pileups
A recent study has shown that R-loops persist at early replicating RDCs in primarily isolated neural stem and progenitor cells[35]. We validated this observation by DNA:RNA immunoprecipitation sequencing (DRIP-

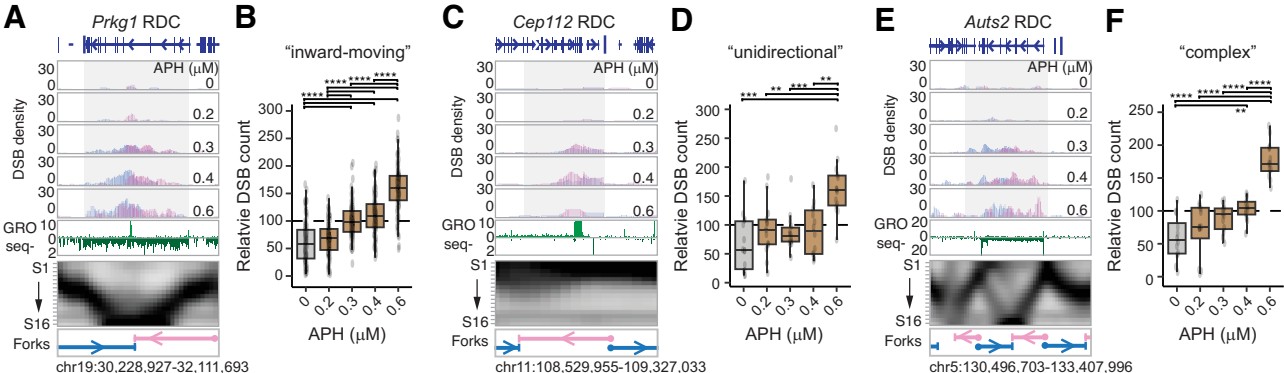

**Fig. 4 | DNA break density at the "inward-moving", "unidirectional", and "complex" RDC follows an APH dosage-dependent kinetic. A, C, E** Multiomics panels provide information on the genomic locus underneath *Prkg1* (**A**), *Cep112* (**C**), and *Auts2* (**E**). The DSB density panels show Dcen and Dtel extended intrachromosomal DSB per 50 kilobases in ES cell-derived NPCs treated with the indicated aphidicolin (APH) concentration. The global run-on sequencing (GRO-seq) panel shows the nascent transcription activity at the genomic loci. The replication heatmaps are shown below, accompanied by annotation for fork directions. **B, D, F** Box plots display the mean and 1.5 quartiles presenting the relative DSB count of the "inward-moving" ($n = 87$) (**B**), "unidirectional" ($n = 15$) (**D**), and "complex" ($n = 12$) (**F**) RDCs. Only the interchromosomal DSB was used to calculate relative DSB density. The upper and lower whisker are the largest and the smallest value no further than 1.5 times of the interquartile range of the hinge. The statistical significance was determined by an unpaired, two-tail Student's $T$-test: *$P < 0.05$, *** $P < 0.01$, ***$P < 0.001$, ****$P < 0.0001$. Source data including exact $p$-values are provided as Source Data file.

seq) assay in ES cell-derived NPCs (Supplementary Fig. 5A). Among the 152 RDCs analyzed, two-thirds of them did not contain R-loops. We found one-third of RDCs contained one to nine R-loops, and only four RDCs (*Ash1l*, *Klhl29*, *Sil1*, *Prkcz*) harbored more than ten R-loops (Supplementary Data 5). In the "outward-moving" *Klhl29* RDC, Dcen and Dtel did not align with the replication fork directions but to the R-loop position. Similarly, the overall DNA break density aligned with the R-loops for the *Sil1*, *Ash1l*, and *Prkcz* RDCs (Supplementary Fig. 5A). The fork directions could not be determined for *Ash1l* and *Prkcz* loci as they were present at the broad initiation zones (Supplementary Fig. 4). This observation suggests that R-loop persistence alters the proportion of Dcen and Dtel, leading to RDCs displaying "overlapping" peaks. In total, the Dcen and Dtel peaks significantly overlapped in 30 RDCs, 22 of which presented at broad initiation zones and contained persisting R-loops (RDC in *Dst*, *Klhl29*, *Trappc9*, *Prkcz*, *Tmem132b*, *Peak1*, *Plekhg1*, RDC-chr9-35.4, *Msi2*, *Slc39a11*, *Cdkal1*, *Zmiz1*, *Samd5*, *Cdkal1*, *Samd5*, *Cep112*, *Csmd2*, *Rere*, *Ptn*, *Ash1l*, *Tln2*, and *Gm12610*; Supplementary Data 5).

To investigate the potential contribution of co-transcriptional transient DNA:RNA hybrids to RDCs, we utilized DNA:RNA immunoprecipitation followed by cDNA conversion and sequencing (DRIPc-seq)[36] in ES cell-derived NPCs. Unlike DRIP-seq, which measures DNA at the R-loop, DRIPc-seq characterizes the RNA sequences associated with DNA:RNA hybrids (Fig. 5B and Supplementary Fig. 5B). As a result, DRIPc-seq reveals the strandness of RNA molecules at the DNA:RNA hybrid hotspots. We found that about 1% of the genome contains coding strand-specific DNA:RNA hybrids. The DNA sequences beneath the strand-specific peaks in DRIPc-seq are rich in GC content (Supplementary Fig. 5C), consistent with the presence of G quadruplex structures on the opposite strand of DNA:RNA hybrids[37,38].

Almost all R-loop-containing RDCs maintain coding strand sequence-specific hybrids (Supplementary Data 5). Interestingly, DRIPc-seq revealed DNA:RNA hybrids not detected in DRIP-seq experiments. *Npas3* and *Grip* genes lacked R-loops (Fig. 5C, E), but in these rightward-transcribing gene loci, most DRIPc-seq peaks aligned with the coding strand (Fig. 5D, F). Similarly, for leftward-transcribing *Ctnna2* and *Tenm3* genes, coding strand-specific RNA sequence enrichment is evident in their respective RDCs (Fig. 5G-K). Notably, 26 RDCs with over ten coding strand-specific DNA:RNA hybrids did not show R-loop persistence using DRIP-seq (Supplementary Data 5). Next, we examined whether dual-strand DNA:RNA hybrids, likely resulting from co-transcriptional transient antisense RNA due to RNA

polymerase II stalling[18], are enriched at RDC (Supplementary Fig. 5D)[19]. As shown for *Ctnna2* and *Grip1* RDC, dual-strand hybrids were enriched in the gene bodies (Fig. 5L, M) and most RDCs (Supplementary Data 5). We next explored whether the density of transcription-coupled DNA:RNA hybrid differs in RDC-containing genes as it is in long genes, as most RDCs are in genes longer than 100 kb while not all long genes contain RDCs[22,23]. We found that the density of the coding strand DNA:RNA hybrids is significantly different between RDC-containing genes versus other long genes (Fig. 5N). In addition, the density of dual-strand DNA:RNA hybrids in RDCs is also significantly higher than in transcribed long genes without RDCs (Fig. 5O). These findings suggest frequent RNA polymerase stalling associated with RDC formation in long genes.

In summary, the DNA break density no longer follows the replication fork directions in RDC at the broad initiation zones and colocalizes with R-loops. In addition, co-transcriptional transient DNA:RNA hybrids are present in almost all RDCs. Lastly, the density of co-transcriptional dual-strand DNA:RNA hybrids are higher in RDC-containing genes than in actively transcribed long genes.

## Head-on collisions correlate to higher DSB densities

Next, we investigated DSB prevalence resulting from head-on versus codirectional collisions at RDCs. At the "inward-moving" RDC at the *Prkg1* locus, we observed a significant increase in the density of DSBs at the head-on fork region, from 15.7 – 28.1 DSB per ten thousand interchromosomal translocations in NPCs treated with 0.3 μM or 0.6 μM aphidicolin (Fig. 4A). In comparison, the codirectional counterpart DNA density decreased from 16.6 to 14.7 DSB per ten thousand interchromosomal translocations under the same conditions (Fig. 4A). At the "unidirectional" RDC at the *Cep112* locus, the head-on Dtel density increased from 1.6 DSB per ten thousand interchromosomal translocation in NPCs treated with 0.3 μM aphidicolin to 4,6 with 0.6 μM aphidicolin treatment. Conversely, the Dcen density remained below one per ten thousand interchromosomal translocations at the codirectional TRC in all aphidicolin-treated conditions (Fig. 4C). We also found that head-on contribution increased at the *Auts2* gene RDC. For these three RDC loci, DNA break density consistently surpassed the codirectional TRC at the head-on TRC (Fig. 4E).

To evaluate head-on and codirectional contributions to RDC DSB distribution at a genome-wide scale, we first assessed all "inward-moving", "unidirectional", and "complex" RDCs. We quantified the proportional changes in Dcen and Dtel within the head-on and co-

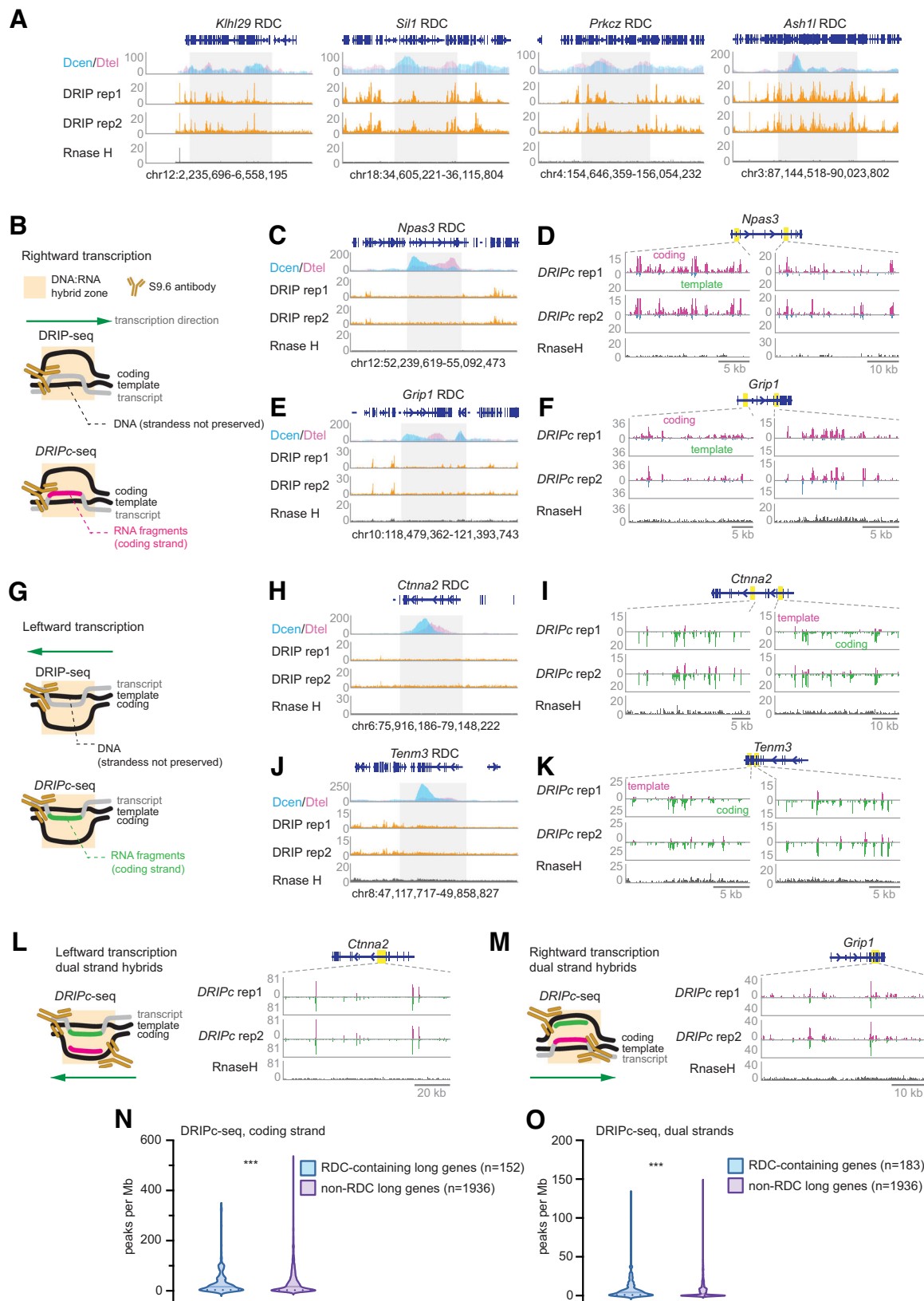

directional regions (Fig. 6A). The balanced scenario—when Dcen and Dtel contribute equally – would yield a 50% DSB proportion for both head-on and co-directional contributions. Significant deviations from this 50% mark suggest differential contributions. Our analysis revealed an increasing DSB contribution from head-on colliding replication forks in a dose-dependent manner upon aphidicolin treatment (proportion of head-on, $P_{HO}$, from 50% to around 60%, Fig. 6B). Conversely, at co-directional replication forks, the DSB proportion (proportion of codirectional, $P_{CD}$) decreased in low-dose aphidicolin-treated NPCs, declining from 50% in untreated samples to ~40% in aphidicolin-treated samples (Fig. 6B), with co-directional DSB contribution remaining consistently low at the same proportion. The

**Fig. 5 | RDCs are Enriched with DNA:RNA hybrids. A** The figures present DSB densities and R-loop levels at four "outward-moving" RDC-containing genomic loci. Centromere-oriented DSB (Dcen) and telomere-oriented DSB (Dtel) density is shown as interchromosomal junctions per 50 kb, while DRIP-seq across repeats and RNase H-treated samples is normalized to 50 million reads. **B** Illustrations of molecules analyzed using DRIP-seq or DRIPc-seq at genomic loci undergo rightward transcription. The top panel depicts a transcription bubble, with a nascent transcript forming a DNA:RNA hybrid. S9.6 monoclonal antibody pools down the hybrid in DRIP-seq, while the bottom panel highlights the same bubble with the nascent RNA molecule in cherry. DRIPc-seq analyzes RNA sequences, preserving RNA molecule strandness. **C, E** Panels depict DSB densities and R-loop levels for the genomic sequence beneath *Npas3* and *Grip1* loci transcribed rightwards, following the organization described in (**A**). **D, F** Panels show DRIPc-seq signals at rightward transcribing genes *Npas3* and *Grip1*, annotating coding and template strands. **G** Illustrations depict molecules analyzed using DRIP-seq or DRIPc-seq at genomic loci undergoing leftward transcription. **H, J** Panels display DSB densities and R-loop level of the genomic sequence underneath *Ctnna2* and *Tenm3* loci that undergo active leftward transcription. **I, K** Panels showing the DRIPc-seq signals at leftward transcribing genes *Ctnna2* and *Tenm3*. Plus signal stands for the template strand, and minus stands for coding at *Ctnna2* and *Tenm3* loci. **L** Left: Illustration depicts the scenario of dual-strand DNA:RNA hybrids at a genomic locus undergoing leftward transcription. Right: DRIPc-seq signals at 100 kb genomic sequences at the *Ctnna2* locus. **M** Left: Illustration depicts the scenario of dual-strand DNA:RNA hybrids at a genomic locus undergo rightward transcription. Right: DRIPc-seq signals at 50 kb genomic sequences at the *Grip1* locus. **N, O** The density of coding strand-only (**N**) and dual-strand (**O**) DRIPc-seq peaks in RDC-containing genes versus in genes longer than 100 kb without RDC. The two-tailed Mann-Whitney test determined statistical significance. The *p*-values are $3 \times 10^{-4}$ (**N**) and $7 \times 10^{-4}$ (**O**). Source data are provided as a Source data file.

disparity between head-on and codirectional contributions was achieved at 20% under the 0.6 µM aphidicolin condition.

Second, we tested whether the difference in DSB density between codirectional and head-on TRCs is due to frequent fork reprogramming in the head-on direction, resulting in slower replication. To assess the relative replication speed (Rs) within RDCs, we compared the number of 50-kb bins in head-on or codirectional TRCs with the number of S-phase fractions spent in these regions (Supplementary Fig. 6A). We found that head-on replication speed was comparable to the codirectional TRC (Supplementary Fig. 6A). To see if DNA break density correlates with relative replication speed, we integrated the absolute interchromosomal DSB density at Dcen or Dtel in each TRC direction (Supplementary Fig. 6B). We observed a weak negative correlation between DNA break density and relative replication speed in genomes with head-on TRC. On the contrary, a slight positive correlation was observed in the codirectional TRCs (Supplementary Fig. 6C). Nevertheless, these correlations were not significant.

Third, we asked if co-transcriptional dual-strand DNA:RNA hybrids correlate to DNA break density. To do so, we stratified "inward-moving", "unidirectional", and "complex" RDCs into three groups according to the number of dual-strand DNA:RNA hybrids detected in them (Fig. 6C). At the 0.3 and 0.4 µM aphidicolin concentrations, DNA break proportions were comparable in RDCs with zero to two transient DNA:RNA hybrids, while the DNA break proportion difference became significantly higher in RDCs with higher hybrid counts.

In summary, the analyses between replication speed and transcription direction demonstrated that head-on conflict significantly elevates DSB density at RDCs, and the elevation is positively correlated with the DNA:RNA hybrid frequency.

## Discussion
### Most RDCs are transcription-associated fragile sites in the timing transition regions
Our findings suggested that RDCs result from the conflict between linear encountering of replication and transcription, while the acting mechanism creating DNA breaks varied. We demonstrated that DNA breaks occurring to the genome underneath the "inward-moving", "unidirectional", and "complex" RDCs exhibit orientations aligning with replication fork directions (Figs. 2, 3, Supplementary Figs. 3, 4) and fork slowing patterns (Fig. 4). In these genomic regions, the DNA break end orientation is in substantial accordance with the TTR direction. TTRs are genomic regions transitioning from early to late replicating domains[39]. We demonstrated that TTRs are hotspots for TRC-mediated DNA breaks. This finding is independent of the DNA breaks at the topological domain boundaries, as those primarily occur at the transition between TTR and early replicating domains[6,39]. In summary, we identified RDCs exclusively containing TTR but not broad CTRs, which we call "TTR Fragile Sites" (TTRFSs, Fig. 7).

It has been suggested that the positioning of TTRs is passively influenced by the location of active IZs[39]. IZs exhibit partial cell type dependence; therefore, the positioning of TTRs could vary across different cell types. We hypothesize that TTRs are not enriched at neuronal gene loci in non-neuronal cells or may only exhibit partial enrichment. In line with this idea, gene enrichment analysis revealed a significant overrepresentation of TTR-containing genes in neuronal functional processes in the mouse NPCs (Supplementary Data 3). Collectively, we speculated that transcription activity at TTR creates NPC-specific RDCs under replication stress.

### A proportion of RDC display CFS or ERFS properties
Twenty-three "inward-moving" and 12 "undefined" RDCs are present at the broad late CTRs (Supplementary Data 5). As the DNA break density increment yet represents a dosage-dependent effect in cells treated with aphidicolin (Fig. 4A, *Prkg1*), we believe these are also DNA breaks resulting from replication stress. Intriguingly, as proposed previously, CTRs are genomic regions where the replication origins are only fired at the late S phase[39]. In RDC containing broad late CTR, Dcen and Dtel density overlap with the CTR (Fig. 2E, G, and Supplementary Fig. 3), suggesting that these DNA breaks primarily occurred at the last S phase fractions. In addition, the high-resolution Repli-seq data indicated that DNA replication is completed at most CTR regions (Figs. 2, 3, Supplementary Figs. 3 and 4) with a few exceptions at the genomics sequences underlying *Magi1*, *Ccser1*, and *Grid2* RDCs, where a gap in the CTR was observed (Supplementary Figs. 3 and 4). This gap is likely due to underreplication at the center of specific RDC-containing genomes. Hence, a subset of broad late CTR-containing RDC may share the DSB-initiation mechanism as CFS[40] (Fig. 7).

For "outward-moving" and RDCs within broad initiation zones, we observed that the DNA break positions are in substantial accordance with the presence of the R-loop (Fig. 5A). We speculate these RDCs share the pathway that creates ERFS[11]. Multiple replication origins are proposed to be simultaneously fired within the initiation zones, leading to "active" DNA replication[39]. At this region, active and frequent origin firing may collide with the R-loop, leading to DNA breaks ahead of the fork. These processes may generate double-ended DNA breaks that are not solitary (Fig. 7). Mechanisms for DSB ahead of the fork were previously proposed by investigating the rDNA genomic in yeast[29]. Nevertheless, we cannot exclude the possibility that the density of active forks at CTR is higher than one per 50 kb, which is below the resolution of our assays.

### Implication of co-transcriptional DNA:RNA hybrids in fork stalling
Our investigation revealed head-on and co-directional conflicts within RDC loci (Fig. 6), mirroring observations reminiscent of R-loop accumulation in certain DNA sequences[19]. Interestingly, despite the scarcity of R-loops detected in most RDC loci, we found that the head-on bias is

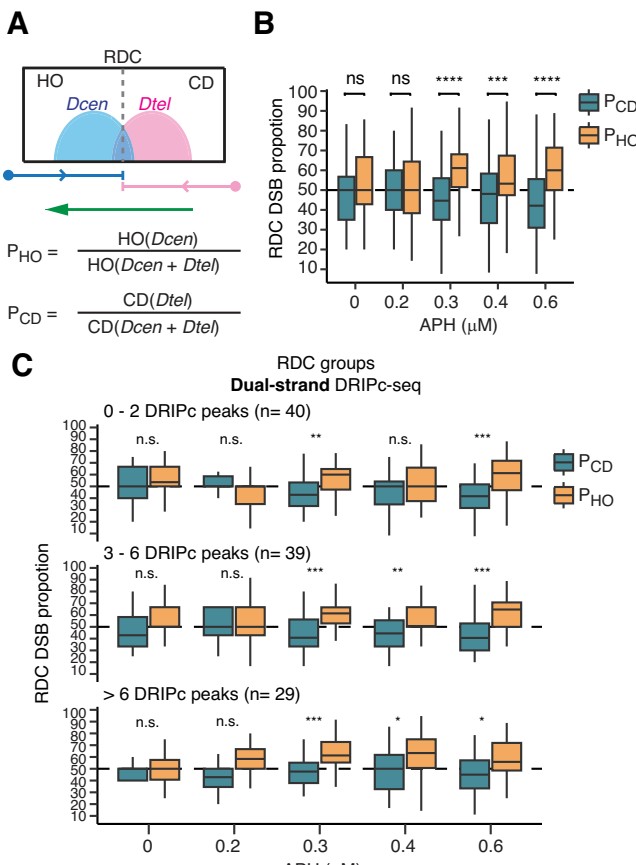

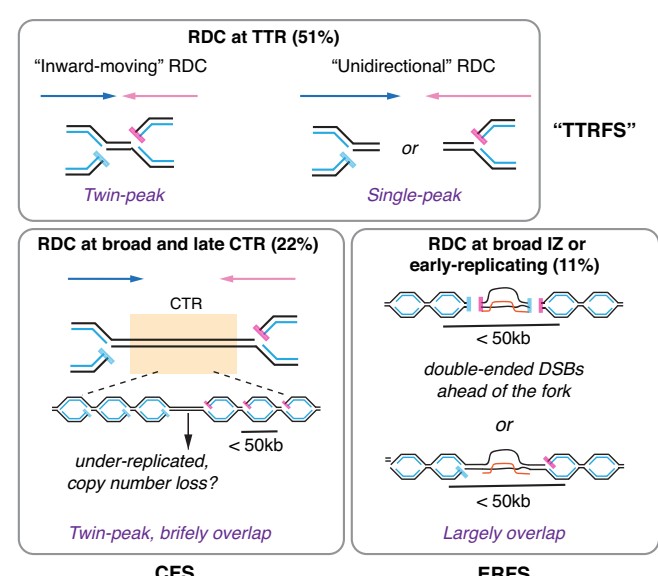

**Fig. 7 | RDC DNA breaks orientation dynamics.** Figure depicts the DNA break position relative to the replication fork at RDCs within the timing transition region (TTR) (top), broad and late constant timing region (CTR) (bottom-left), and broad initiation zone (IZ) (bottom-right). TTRFS: timing transition region fragile site. CFS common fragile site. ERFS early replicating fragile site.

**Fig. 6 | Head-on TRC creates higher DNA Breaks at RDCs. A** Scheme explaining proportions of head-on and co-directional DSBs. The upper panel shows a schematic example of an RDC. DSB density for centromere-oriented DSB (Dcen) and telomere-oriented DSB (Dtel) is shown in light blue and pink, respectively. Blue and pink arrows indicate replacement forks moving rightward or leftward, respectively. The green arrow indicates the transcription direction. A dashed line within the RDC box indicates the termination zone. HO head-on collision. CD co-directional collision. The lower panel explains the calculation of the proportions of DSBs from Dcen and Dtel in the head-on orientation ($P_{HO}$) or the co-directional orientation ($P_{CD}$). **B** A box plot displays the mean and 1.5 quartiles of DSB density proportions within the head-on and co-directional compartments of all "inward-moving", "unidirectional", and "complex" RDCs ($n = 114$). A horizontal dash line signifies where Dcen and Dtel contribute equally to the $P_{HO}$ or $P_{CD}$ compartment (50%). The upper and lower whisker are the largest and the smallest value no further than 1.5 times of the interquartile range of the hinge. Statistical significance was determined through a nonpaired, two-tailed Student's T-test, with significance levels indicated as follows: $^*P < 0.05$, $^{**}P < 0.01$, $^{***}P < 0.001$, $^{****}P < 0.0001$; ns: insignificant. APH aphidicolin. The exact $p$-values are, from lowest to highest APH concentration: $1.8 \times 10^{-1}$, $1.4 \times 10^{-1}$, $1.4 \times 10^{-8}$, $5.5 \times 10^{-4}$, and $8.8 \times 10^{-8}$. **C** Box plots display the mean and 1.5 quartiles presenting DSB density proportions of the HO and CD RDCs in correlation with different dosages of APH treatment. The upper and lower whisker are the largest and the smallest value no further than 1.5 times of the interquartile range of the hinge. RDCs were stratified into three groups according to the number of dual-strand DRIPc-seq peaks. Statistical power was determined as described in (**B**). Source data including exact $p$-values are provided as Source Data file.

attributed to co-transcriptional DNA-RNA hybrids (Fig. 6). An example of such hybrids includes co-transcriptional R-loops formed between the nascent transcript and template DNA, which are nascent transcript reanneals to the template DNA[18]. Similarly, the abundant DNA:RNA hybrids formed by short RNA primers during DNA replication, notably on replication forks, are swiftly resolved during the maturation of Okazaki fragments[41]. In addition, the abundance of dual-strand DNA:RNA hybrids is significantly higher in genes that contain RDCs

than in genes >100 kb that do not present as RDC (Fig. 5O), suggesting RNA polymerase pausing may play a role in licensing RDC formation.

## DSB end orientation at the "inward-moving" RDC implies a mechanism promoting copy number loss

We speculate that DNA breaks at "inward-moving" RDCs may foster deletions (Fig. 7). In cases of "inward-moving" RDCs flanked by two TTR, DSB ends at the rightward-moving fork pointing to the DSB ends at the leftward-moving TTR (Figs. 2, 3 and Supplementary Figs. 3 and 4). We propose that this head-to-head DSB orientation within *cis* configurations may foster mechanisms reliant on homology or end joining, potentially leading to deletions. For instance, at the *Npas3* locus, a 700 kb intragenic interval exists between Dcen and Dtel peaks. Fusing one Dcen and one Dtel within the *Npas3* gene could result in a critical exon deletion, a genomic alteration strongly associated with conditions like autistic spectrum disorder, schizophrenia, and glioblastomas in humans[42]. Such deletions could emerge at a somatic level during replication stress in NPCs. Given that a significant proportion of RDC-containing genes govern cell adhesion or synaptogenesis[22,23], deletions in NPCs could reverberate to daughter neurons, thereby impacting neuronal function.

Exploring the linear interplay between transcription and replication unveils several research frontiers. It remains to be systematically explored whether TTR, active transcription, or the simultaneous presence of both factors mediate RDC formation. Furthermore, the role of Dcen and Dtel in facilitating genomic rearrangements, specifically those involving exon exclusion on the neuronal gene template, remains to be resolved. These endeavors will be pursued using systematically developed genetic tools expected to emerge soon.

## Methods
### Cell culture
We used mouse ES cell-derived NPC cell lines that were Xrcc4-/-p53-/-, clone Nxp010, for the described experiments. The Xrcc4/p53-deficient mouse embryonic stem cells were cultured in DMEM medium supplemented with 15% ES cell-grade fetal bovine serum, 20 mM HEPES, non-essential amino acids, 100 U/ml Pen/Strep and glutamine mixture, 0.1 mM beta-mercaptoethanol and 1000 U/mL ESGRO recombinant

mouse leukemia inhibitory factor (LIF) on a monolayer of confluent irradiated mouse fibroblasts. To differentiate ES cells into NPCs, ES cells were plated on laminin/poly-L-ornithine-coated plates and cultured in N2B27 medium (50% DMEM/F12, 50% NeuralBasal, 1% modified N2 supplement, 2% B27 supplement without RA, 1X Glutamax) for 7 days. Cells were then passaged to laminin-coated plates and culture in NBBG medium (NeralBasalA, 2% B27 supplement without RA, 0.5 mM Glutamax, 10 ng/ml human EGF, 10 ng/ml mouse FGFb) for another five to 6 days. For LAM-HTGTS experiments, ES cell-derived NPC cells were treated with or without indicated aphidicolin concentration for 72 h, and the aphidicolin concentration was further reduced to half for another 24 h. Reagents used in cell culture are listed in Supplementary Data 6.

## High-resolution replication sequencing

The high-resolution, 16 fractions Repli-seq were generated and analyzed as described in the Supplementary Methods and in previous publication[26]. In brief, 20 million ES cell-derived NPC were incubated with 400 μM BrdU for 30 min in the case of untreated samples, while this duration was extended to 45 min for APH-treated NPCs. BrdU-labeled cells were fixed in 70% ethanol and stained with propidium iodide. Cells were sorted by BD FACSAria Fusion based on DNA content. Library preparation details are described under the Supplementary Methods section. A 50-kilobase genomic bin was selected based on the assumption of a fork speed of 1.8 kb/min. With 30 min of BrdU labeling, this allows the incorporation of the analog in at least 50 kb DNA per fork. The aligned reads were normalized, and Gaussian smoothed as previously described[26]. Reagents used for Repli-seq are listed in Supplementary Data 6.

## Defining DNA replication features

The matrices smoothing, scaling, and BIRCH clustering of the data have been performed, using 80 clusters for all conditions/replicates (the Python code is available at github.com/ClaireMarchal/High-Res_repli-seq_Features) as described[26]. The cluster sorting and attribution to each locus has been slightly modified compared to Zhao et al[26] as the original method was missing many non-IZs features in the XRCC4/p53-deficient NPC datasets: the maximum centroid fraction of each Birch cluster has been used to attribute a corrected fraction (1 – 16) to each genomic locus (instead of attributing a birch cluster 1–80 as in[26]). These corrected fractions were then used to call features described before[26] with a slight modification for the TTR call, for which steps with three consecutive bins belonging to the same fraction were tolerated within a TTR.

## Termination meeting point prediction

We employed a convolutional neural network to determine the termination meeting points from the available 16-fraction Repli-seq datasets ([26] and GSE137764), leveraging the insights it acquired from the OK-seq datasets.

The network comprises three distinct phases. As fork direction is determined by connecting an IZ to its nearest TZ, the initial phase involves identifying genomic regions that contain defined IZ and TZ features. This step was exclusively conducted utilizing OK-seq datasets generated using mouse ES cells (ref. [43] and GSM3290342). To predict the TZs, we applied the OKseqHMM R package[44] to the mouse ES cell OK-seq dataset. To determine initiation zones, we fitted a LOESS model[43] to replication fork directionality (RFD—a metric used in OK-seq paper) values from mouse ES cell OK-seq data. IZs flanked by two TZs with RFD values crossing zero (going from negative values to positives) were preserved as reference IZs.

The second phase involves generating training data. Initially, we select training regions encompassing at least one IZ, with a neighboring TZ situated nearby, to ensure the region contains a complete replication fork direction. Notably, we observed instances where

replication fork directions extended beyond 1.5 Mb in length in various areas. Consequently, the training area must exceed 1.5 Mb in size. To address this, we opted to enhance the dataset by introducing additional training samples produced from downscaled Repli-Seq and IZ data by a factor of two along the genomic axis, merging two adjacent 50 k bins into 100 k bins. Subsequently, we configured the training data to comprise 30 bins each, effectively spanning 3 Mb in each training area. This adjustment extended the coverage beyond the 1.5 Mb threshold.

Following this, the entire genome was partitioned into 1.5 Mb regions (30 bins of 50 Kb each) and, more significantly, three Mb regions (30 bins of 100 Kb each), and data from both segmentation strategies were unified into a cohesive dataset. This approach increased the total amount of training data and allowed for the covering of longer replication forks. Overall, 2806 training samples were generated using a 0 x 24 stride. This approach created training data that shared six bins with the upstream and downstream regions and 24 bins that were not shared. The reason for the overlap is that only 24 non-shared bins were predicted when the network was trained, but the network can still make decisions based on data beyond that region (+/- 6 bins). With this, we predict only non-overlapping regions but look at neighboring data. These samples included 763 instances with a single termination zone, 633 with two termination zones, 209 with three termination zones, 30 with four termination zones, and six with five termination zones.

The third phase is to train the model with defined training data. We choose a fully convolutional deep learning model to handle varying genome sizes. To enhance sensitivity to various scales of replication forks, we utilized single inception module[45] with dimension reduction (5 x 5, 7 x 5, 9 x 5 filters and max-pool layer) followed by three convolutions with 1 x 1, 1 x 1, and 1 x 9 filters to bring dimensions under IZ labels. The training was executed using Adam optimizer to minimize the Dice loss function. A mini-batch of size 32 was used with a learning rate of 0.001 and a dropout of 90%. Activation functions within each layer were the scaled exponential linear unit (SELU) activation function, and the final output layer was a sigmoid activation function. To assess the model's performance, a 10-fold cross-validation was employed. It was implemented using the Keras and Tensorflow packages within the R programming environment. Further details and the associated code are accessible at DOI: 10.5281/zenodo.10832658.

This method compensates for the BIRCH algorithm, which does not recognize TZ broader than 100 kb. The neural network was trained using diverse features extracted from published mouse ES cell Okazaki-sequencing (OK-seq) datasets and high-resolution Repli-seq datasets generated in wild-type F121/CAST NPCs[26,46]. Post-training, this model achieved an 80% recall rate with a precision exceeding 80% within a 50 kb window size. We then applied the trained convolutional neural network model to high-resolution Repli-seq data. Subsequently, the direction of the replication forks was determined by connecting the initiation zones to the closest termination meeting points. Forks shorter than the Repli-seq bin size (50 kb) could not be determined. We considered only the consistent fork directions across technical repeats.

## LAM-HTGTS

Libraries were prepared following established protocols[22,23]. Reads from demultiplexed FASTQ files were aligned to the mm10/GRCm38 genome assembly using Bowtie2 and further processed via the HTGTS pipeline (https://github.com/brainbreaks/HTGTS). To induce bait on chromosomes 5, 6, 8, 12, or 17 in Xrcc4-/-p53-/- ES cell-derived NPCs, we nucleofected five million cells with five μg of spCas9/sgRNA-expression plasmids (pX330-U6-Chimeric-BB-CBh-hSpCas9, Addgene #42230) by Nucleofector 2b. Specific sgRNA sequences targeting bait locations were cloned into separate plasmids accordingly. Cells were collected 96 h post-nucleofection for genomic DNA extraction and subsequent

LAM-HTGTS analysis. Reagents and oligos used for LAM-HTGTS are listed in Supplementary Data 6.

## RDC calling

Libraries used in this article, off-target calling, and data clean-up process are described under Supplementary Methods. Only DSB detected at the non-viewpoint chromosome are subjected to statistical analyses and plotting in Figs. 2, 3, 4, 5, 6, and Supplementary Figs. 3, 4, 5, 6. We excluded bait viewpoint-chromosome for analyses as the Dcen and Dtel recovery rate is unbalanced. The bait preferentially recovers 15–25% more downstream DSBs at the break site chromosome than the upstream. Using bait viewpoint chromosome DSB resulted in an overrepresentation of the centromeric DSB end when the bait had a centromeric orientation. The bait with a telomeric direction resulted in an overrepresentation of the telomeric DSB end. The bias due to bait DSB end orientation on the bait viewpoint chromosome was as significant as 20%. DSB end recovery bias was not present on the non-viewpoint chromosome.

For peak calling, we extended LAM-HTGTS junctions by 50 Kb symmetrically in both directions, and pileup islands were determined for telomeric-only (Dtel), centromeric-only (Dcen), and all junction orientations (Dtel + Dcen). A negative binomial model for estimating the expected pileup value for each chromosome/condition/junction-orientation triplet was derived, and a p-value was calculated for each pileup value concerning model expectation. Regions with a p-value below 0.01 joined (maximal gap 10 Kb) to create seeds. These seeds were further joined with other seeds (maximal gap 100 Kb) to form islands. Islands are extended up and downstream to include regions below 0.1 significance. Overlapping orientation-specific islands are further joined to form an initial RDC list that is further filtered to contain at least 100 Kb below 0.01 p-value and be of at least 300 Kb in length when considering extended regions. The broadest range of all overlapping and significant islands determined RDC. The RDC-calling algorithm is available at https://doi.org/10.5281/zenodo.10832658. The algorithm used in this manuscript aimed to define orientation-specific islands and join islands to form RDC. The algorithm called 28 RDCs from previously published datasets[22,23], and 143 RDCs from the LAM-HTGTS datasets generated in this manuscript. All 28 RDCs called using the orientation-specific algorithm are previously defined RDCs. Due to the smaller library sizes (-10 k per experiment) in the previously published datasets, the newly generated libraries (-30 k per experiment) contributed to most of the RDC analyzed in this manuscript. We annotated whether the RDC is additionally identified or described previously in a column in Supplementary Data 2. The additional RDC identified by this combinatory approach is contributed by enhanced data depth, as the additional RDCs already display slightly enhanced DNA break density in the previously published datasets. Gene ontology (GO) enrichment result is ordered by a multiple testing approach FWER (family-wise error rate for over-representing genes)[47]. GO terms with a P-value smaller than -log10 were reported in Supplementary Data 3.

## LAM-HTGTS Library normalization

To maintain the phenotype wherein increased replication stress leads to more DNA breaks, we standardized the library size to establish off-target sites where DSB frequency remains unaltered by aphidicolin concentration for each utilized bait in aphidicolin-dosage experiments. The off-target sites employed for normalization include chr2:165364620-165369885 for the chr5 bait, chr4:141559295-141560757 for the chr6 bait, chr14:22764896-22769859 for the chr8 bait, and chr12:111040435-111040503 for the chr12 bait. DSB counts from varying concentrations were adjusted based on the off-target weighting. The "off-target-normalized libraries" were utilized to determine relative DSB enrichment.

To standardize DSB counts between RDCs (Figs. 4, and 6), the DSBs identified in untreated, 0.2, 0.3, 0.4, and 0.6 µM aphidicolin-treated cells were aggregated for each RDC (ΣRDC). ΣRDC was divided by five to calculate the mean DSB count. The value for each RDC illustrated in Fig. 2D was determined using the equation below. Only DSBs identified through inter-chromosomal bait were considered for analysis in this context. The relative DSB count displayed in Fig. 4 represents $\log2[P_{RDC}/(\Sigma RDC/5)]$, where $P_{RDC}$ signifies DSB counts per treatment condition per RDC.

To calculate TRC break density (Supplementary Fig. 6C), genomic 50 bins encompassing actively transcribed genes were analyzed. The individual bin was connected into one TRC if the same gene encompassed them. The DNA break density was displayed as inter-chromosomal translocation per 50 kb.

## GRO-seq

GRO-seq libraries were prepared as previously described[22,24]. For each GRO-seq experiment, 5 – 10 million ES cell-derived NPC nuclei were isolated for global run-on analyses. We extracted total RNA with the Trizol (Ambion, 15596018), and the BrdU-incorporated RNA was enriched with the agarose-conjugated anti-BrdU antibody (Santa Cruz, sc-32323, dilution 1:170). At least two technical replicates per experiment condition were performed. GRO-seq FASTQ files were aligned to the genome build mm10/GRCm38 through Bowtie2 and processed as described[22]. We set a cutoff value of 0.05 reads per kilobase per million (RKPM) to determine gene transcription activity. GRO-seq libraries for XRCC4/p53-deficient NPC are deposited under GSE233842. Reagents used in GRO-seq experiments are listed in Supplementary Data 6.

## DRIP-seq and DRIPc-seq

ES cell-derived NPCs treated with DMSO were used for DRIP-seq and DRIPc-seq experiments. We follow the published protocol[36] with minor modifications described under Supplementary Methods. Two independent experiments for RNaseH-treated or untreated samples were performed. Significant DRIP- or DRIPc-seq signal enrichment was determined by MACS2, as described in the Supplementary Methods. Reagents used for DRIP- and DRIPc-seq experiments are listed in Supplementary Data 6.

## Reporting summary

Further information on research design is available in the Nature Portfolio Reporting Summary linked to this article.

# Data availability

The raw and processed LAM-HTGT and GRO-seq data generated in this study have been deposited in the GEO database under accession code GSE233842. The raw and processed Repli-seq, DRIP-seq and DRIPc-seq data are available under accession code GSE254765. The corresponding location for data generated in this study are provided in the Supplementary Data 1. In addition, the published LAM-HTGTS data used in this study are available in the GEO database under accession codes GSE106822 and GSE74356. The Okazaki-sequencing data are available in the GEO database under accession code GSM3290342, and the CAST/F121-9 Repli-seq data used for convolution network training are available in the GEO database under accession code GSM137764. All sequences were mapped to the mouse genome mm10 (https://hgdownload.soe.ucsc.edu/goldenPath/mm10/bigZips/). Source data are provided with this paper.

# Code availability

Codes and bioinformatic analysis pipelines were available at Brain-Breaks GitHub (https://github.com/orgs/brainbreaks/repositories). Specifically, the data were analyzed by using the following packages: RDC calling and fork direction prediction (https://doi.org/10.5281/zenodo.10832658), LAM-HTGTS pipeline (https://doi.org/10.5281/

zenodo.10843397), GRO-seq (DOI: 10.5281/zenodo.10838367), and high resolution Repli-seq (https://doi.org/10.5281/zenodo.10838365). The link to package used for each assay is summarized in Supplementary Data 1.

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

## Acknowledgements

This work is supported by the Helmholtz Young Investigator grant and an ERC starting grant BrainBreaks to P-C W and NIH grant R01CA270335 and GM083337 to D. M. G.. V. S. I. acknowledges the DKFZ international Ph.D. program for a travel grant. B.D. received support from a Chinese Science Council scholarship. An Erasmus scholarship and a scholarship from Bologna University supported L. C. We extend our appreciation to Boston Bio Edit, Michael Dill, and Duncan Odom for their valuable language editing contributions. We also express our gratitude to the members of the Wei and Gilbert labs for their insightful discussions and to the Cell Sorting and NGS core facilities at the German Cancer Research Center (DKFZ) for their exceptional services.

## Author contributions

L.C., V.S.I and P.-C. W. designed the research. L.C., V.S.I., M. G., G. D. M., A.-M., J. H., B. D. and M. A. performed experiments. L.-C.W. and S. A. designed the computational analytic pipelines. L.C., V.S.I., P.-C. W. and L.-C. W. analyzed the data; P.-C. W wrote the paper. L.C., V.S.I., G. D. M., B. D., P.-C. W., A.V and D. M. G. helped polish the paper.

## Funding

## Competing interests

The authors declare no competing interest.
