## [Peer Review File · Nature Communications]

Linear Interaction Between Replication and Transcription Shapes DNA Break Dynamics at Recurrent DNA Break ClustersEditorial Note: Parts of this Peer Review File have been redacted as indicated to remove third-party material where no permission to publish could be obtained.

REVIEWER COMMENTS

Reviewer #1 (Remarks to the Author):

The authors of Ionasz and Corazzi et al used Repli-seq and OK-seq datasets to derive by neural network training, replication termination zones that would link with replication initiation zones to correlate and explain the formation of chromosome orientation-specific solitary DSBs in previously identified transcriptionally active recurrent break cluster (RDC) genes of ES cell derived neural progenitor cells deficient in Xrcc4 and p53. The authors then designate each RDC 1 of 5 categories and highlight their most distinguishing features. Using DRIP-seq, they demonstrate that the replication-dependent RDCs formed do not appear to be affected by paused transcription and subsequent transient RNA:DNA hybridization that would constitute an R-Loop. Notably, great effort was put forth to demonstrate differential effects of transcription on the formation of RDCs, which varied in significance by locus. Thus, the authors identified multiple factors driving orientation-specific DSB formation that include replication fork direction, replication timing, and locus-specific effects of transcription on RDC activity, which for the latter two parts speculatively implicates local chromatin differences according to cell type as a contributing factor to the overall RDC generation model. Overall the work provides compelling evidence that TRCs are causal to the formation of many RDCs but that the slowed replication fork DSB generation is affected by more than just head-on transcription for some RDCs.

Major points

Although the DSB directionality matches well with replication directionality as described, there are two instances where statements made suggest either an RDC category behaves in a common fashion for like RDCs in that category, rather than just one RDC displaying this phenotype from the data shown, or an alternative interpretation of the data shown could be made.

First, as described in the abstract "This pattern, however, reverses in early-replicating DNA" and on page 4 lines 8-9: "We found that the orientation of DSB ends outward moving fork directions were opposite the proposed model", it is clear from Fig2 F one example shown looks to have red/blue twinning rather than the blue/red twinning seen for inward fork DSBs but the 2nd example, March 1, does not have the red/blue twinning and looks more similar to the inward pattern. Can the authors either firm up this opposite DSB pattern seen for the outward moving category with other RDCs? Otherwise the statements made really only apply to one locus of that RDC category and become less significant.

Second, the Large gene RDC is listed as unidirectional (rightward) in Fig 2D based on RDC designation, but data in Fig 3, which look reproducible to Fig 2 data, show a pink late replicating leftward peak for Large (Fig 3C) just outside of the early replicating RDC designated area that increases with ATRi; clearly this does not connect RDC seeds according to described calling parameters but this could also be categorized as an inward fork DSB and thus implies more than one category may satisfy an RDC or that the RDC calling for this locus is not accurately depicting the translocation data presented. Furthermore, to support the point made for early replication sites firing dormant origins with ATRi in Fig 3B, it could be argued that the shoulder of the blue peak in Fig 3C is shifted in a similar fashion. This was not noted in the manuscript and may need to be described more fully in the results.

Given the value in learning about which orientation DSBs are formed at TRCs, the authors should discuss further whether the ends presented are from the leading or lagging strand or if it is not clear.

Related to this point, figure 2 diagrams were helpful but seemingly showed both strands of each fork as broken, which would not be solitary.

Although the first half of the manuscript was devoted to categorizing RDC subsets based on replication fork progression, it was not clear whether replication fork progression status in the RDC (i.e. DSB position relative to initiation/termination) itself plays a greater role in determining the extent to which transcription is a significant causal mechanism. For instance *Ctnna2* and *Foxp1* had opposing results despite both being inward moving (late Rep) DSBs. *Nrxn1* is listed as complex in Table S1, mid-S phase in Fig S6 and *Ptn* goes from late in ES cells to mid-early in NPCs which should translate to unidirectional/outward for the latter. Connecting these points, perhaps, somewhere in the last 2 paragraphs of the discussion might be useful to make the manuscript more cohesive overall.

Can the authors speculate on what twin versus overlapping peaks may imply with regard to TRCs of inward or outward moving forks?

The reader may be better prepared to understand outward vs inward moving contexts by explicitly connecting them to early replication and late replication; logic would then indicate unidirectional would then be mid-S phase replication. However, if there are examples of inward/early or the opposite, then those conditions should be discussed as then you would have converging replication in early replicating genes.

Table S1 should include some type of NPC replication timing designation for each RDC especially if there are exceptions to the above comment.

The DRIP-seq analysis describes RDCs as having minimal R-loop accumulation, consistent with an earlier publication, and in RDC genes that are responsive to forming DSBs from transcription alterations, suggesting that replication-driven RDC effects are not necessarily due to transcriptional pausing that would promote DNA:RNA hybridization. However, no mention was made to describe any positive R-loop peaks from early replicating genes to demonstrate some level of rigor for a negative result. Furthermore, speculation is presented in the discussion for even more transient or different (dual strand) DNA:RNA hybrids contributing to the mechanism despite the negative correlation of DRIP-seq peaks and RDCs. It would be helpful if the authors can clarify this discrepancy in terms of R-Loop contribution to RDCs. Perhaps the DRIP-seq findings could be complemented with strand specificity of DRIP-seq peaks using the same protocol publication (DRIPc-seq) that is cited in the methods or use nuclear DRIP-seq to potentially support the conclusion and discussion point.

There is a materials section detailing the various sequencing methods used but this reviewer could not find data deposition details for GRO-seq (both ES and NPCs) and DRIP-seq in the manuscript but found one GRO-seq (ES Cast.129 cells) and DRIP-seq in the two indicated GSEs. NPC GRO-seq, if done previously, should be referenced with the GSE number.

While it is appreciated which newly generated baits were included for LAM-HTGTS, some quantitative value (#s or %) of what is newly added relative to what was combined with previous studies or which figures would be completely new would help benefit readership understanding of what is added and clarify how panels covering the same loci in prior publications are different.

Page 8 lines 11-13: Provide a reference or additional RDCs indicating the early but not mid/late replicating sites fire dormant origins with ATRi. This could be added to Table S1 for RDCs with enough power.

Page 15 line 2: There is no Figure S5E in this version of the manuscript and either this data should be included, given that the reference provides evidence that contrasts with prior observations seen in DT40 and human cancer cells, or the discussion point should be removed.

Minor points

Generally, please read through carefully for grammatical errors and incorrect figure panel citations.

Page 2 line 5 became → becomes

Page 5 line 20 viewpoint → bait viewpoint

Page 5 lines 24-27 needs more clarification and might be better described as it relates to within each RDC rather than the genome as translocations to sites outside of RDCs may have other mechanisms in play.

Page 8 line 11: Capitalize new sentence

Page 8 line 18: DTM is not clear, per thousand interchr. DSBs, bps, or reads?

Page 10 line 11 R-loop presence was absent → R-loops were not detected

Page 14 line 6 MUS81 are → is

Page 23 lines 20, 23, 28: it appears the wrong figures are being referenced.

Page 24 line 1: Fig3C is not the correct figure reference

Fig. 2B legend: not clear what the units of DSB density are: DSBs/100kb as an absolute number or percentage or some type of 100kb sliding window? Absolute RDC numbers are not reported so it is difficult to evaluate which RDC patterns are more significant.

Fig. 4C legend should indicate how many RDCs were used as was indicated for Fig 3E; also the 4C panel is difficult to view the stripes with the color scheme shown as they both look like solid colors at low paper viewing magnification; the panel legend references two conditions but the graph has three colors (DMSO). This is not clear but an otherwise excellent panel from a scientific viewpoint.

Figures 5C,D and 6C,D should have some pictorial indication in the panel as to which are aph treated.

Fig. 6 legend (D) describes a figure 4D panel that does not exist in this version.

Fig. 7A not clear what the extra red circle cross is indicating below mPGK

Fig 7B,C,E,F should indicate the cell line and genotype used since this is different from other main figures that were NPC; Xrcc4-/-p53-/-.

Fig 7C library numbers should also be reported in the legend to be in alignment with Fig 7E,F

Fig. S5B is missing description of promoter colors

Reviewer #2 (Remarks to the Author):

The Ionasz and colleagues are studying a very interesting and fundamental phenomenon of replication-transcription collisions in mammalian cells. I generally find the data they generated very interesting and studying directly dose-dependence of eg aphidicolin treatment or RTC is very interesting and deepens our understanding of mechanisms of RTC, as well as studies of RTC at isoforms or the impact of R-loops.

The paper have several weaknesses though, one of the significant. The authors consider that their main result and "ground breaking discovery" is that they are able to detect 1-ended DSBs in the sequencing data. However, detection of 1-DSBs in sequencing data was reported several years ago and used to deducing local directions of replication forks (Zhu et al. 2019, Fig. 5c, attached) (<https://doi.org/10.1038/s41467-019-10332-8>). I am also very confused by the "model" that the directionality of the collapsed fork will be reflected in the sequenced read and a sentence in the abstract saying that this "model" is not correct in early replication. The relationship between fork direction and to which strand the resulting DSB would map is rather straightforward and is explained in more detail in Zhu at al 2017 (<https://doi.org/10.1101/171439>) (Fig. attached). If the authors reach different conclusions it can be caused by their predictions of local fork direction being inaccurate or by coexisting fork directions and frequencies of one-ended DSBs not being a linear function of frequency of forks traveling in different directions (since the authors and others before concluded that heads-on collisions are more likely to result in DSBs).

"However, the exact orientation of DSB ends at the replication fork within living cells remains an unanswered question."

This stamens is also not correct due to the reports in Zhu at al 2019.

Another issue is that state of the art is not that well explained in abstract and introduction. The information on the R-loops is very basic and not up to date. It is true that a 2016 paper reported seeing R-loops only in 2-15% of the transcribed genome, but with technique improvement it has likely changed and is not a fundamental R-loop characteristic.

On the contrary, more recent findings on the R-loops could be added, for example emerging understanding that many R-loops arise physiologically and do not promote RTC or DSBs, unlike some pathological (or toxic) loops (Promonet et al. 2020 <https://doi.org/10.1038/s41467-020-17858-2>). More interesting fact that R-loops are typically enriched at TSS and TTS but can also form inside the genes (e.g. Fig. 6 in Promonet et al. (2020), <https://doi.org/10.1038/s41467-020-17858-2>). Moreover, Promonet et al. reported that many R-loops are physiological and only subset of them is causing DSBs in a context dependent manner.)

Language of the paper is often confusing and would benefit from simplifying.

10-11 "These DSB ends possess inherent orientations, attaching themselves to either centromeric or telomeric sequences on mammalian chromosomes"

I understand what authors are trying to say here, but talking about DSBs "attaching themselves" to either centromeric or telomeric sequences creates confusing visual. Since those DSBs result from broken forks, they simply inherit those forks' orientation.

"As replication stress intensifies, excessive DSBs at forks became sensitive for rearrangements". How do we know which breaks are "excessive"?

"Gaining an understanding of these orientations holds the potential to illuminate the processes of genome rearrangements under conditions of replication stress." I understand it boils down to knowing whether DSB originates from HO or collinear collision would help to understand mechanism of genomic rearrangements? If so, an example would be helpful.

"Notably, termination zones exhibited a higher degree of dynamism compared to initiation zones."

What is a higher degree of dynamism? And both initiation and termination zones are just genomic intervals, do author mean that termination zones, as expected, would be more cell-line and condition dependent? if so, stating it more clearly would be helpful.

"We proposed that fork stalling at inward-moving forks yields centromeric DSB ends at right-moving forks and telomeric DNA ends at left-moving forks (Fig. 2A)."

Until this, DSB direction were always described as either (from) centromere or (from) telomere, but now it is mixed with "right" forks, making this sentence difficult to understand.

In terms of purely stylistic remarks, "fork directions" cannot "extend to 1.5Mb", DSBs are also not a subject of rearrangements, genome is.

Reviewer #3 (Remarks to the Author):

General comment:

Using LAM-HTGTS, a technique mapping simultaneously DSB and their orientation, coupled with a CRISPR-Cas9 inducible system in mouse neural progenitor cells submitted to replication stress, Ionasz et al. propose that the orientation of DNA replication directs the orientation of DSB end in respect to centromeres and telomeres. In a second part of the study, the authors investigate the contribution of transcription and transcription-replication conflicts (TRC) in the occurrence of DSB at specific loci prone to generate clusters of DNA breaks (termed RDC). Both aspects are of general interests to understand how DNA damage and replication stress alter genome stability.

Although the first part of the study provides a well documented description of DSB density and orientations according to the directionality of DNA replication, the data presented in the manuscript regarding the impact of transcription and TRC on the occurrence of DSB are, in my opinion, much less substantiated (see major issues). Indeed, in most of the cases/loci studied by the authors, no strong effect of transcription on RDC occurrence is seen. The authors - and I agree with them - even point out in the discussion that there must be another key factor determining RDC occurrence that is not transcription per se. Hence, I found that conclusions drawn in the second part of the manuscript are not well supported by data and are limited to a few loci that are not behaving similarly. In my opinion, the authors either need to perform a series of experiment to identify what is contributing to RDC

occurrence - but it might be out of reach in the context of revision - or alternatively, they could refocus the manuscript on the first part of the paper linking DNA replication directionality with DSB ends orientation.

Major issues:

- Figure 4 and Pages 9-10, lines 15-28 + 1-3. The authors conclude that Head on - TRC increases DSB density, in a dose-dependent manner, under APH treatment by analyzing 85 RDC. In the same graph, they also show that Codirectional - TRC decreases DSB density under APH treatment, in a manner that is not proportional to the dose of APH. Although supporting data look convincing, the authors need to formulate hypothesis and to explain these results. For example: Does the APH treatment slow down DNA replication in a way that the replisome never reaches the transcription site? If this is the case, why would it only be the case for the CD - TRC? The authors should provide data and/or further analyses to elucidate that point.

- The authors conduct a series of experiments and analyses to identify the determinants of DSB occurrence in RDC regions, specifically in respect to transcription and TRC. One of the major issues of the manuscript is that most of the time the authors try to deduce general features from the analysis of very few (if not a single!) loci. For instance, in Figure 3, the authors analyzed a single representative gene for early replicating and mid-replicating regions and draw general conclusions on the differences between early and mid/late replicating regions. This bias becomes even more problematic later on, starting from Figure 5, where the authors argue that transcription is required for RDC occurrence in *Cttna2* - which seems to be genuinely the case - but in the same figure they also show that it is not the case for another locus (*Foxp1*). Although I agree that at this stage in the manuscript they specifically conclude on the effect of transcription activity in RDC formation at the *Cttna2*, later on in Figure 7, they show that DSB occurrence at the same locus is not proportional to transcription level. Then, in Figure 6, the authors now conclude that it is the full-length transcription that is required for RDC occurrence in another locus (*Nrxn1*). Why are these 3 loci behaving differently? Can we assert a general rule for RDC occurrence? Unfortunately, with the data presented, I don't think so. In the last figure, the authors demonstrate that activating transcription at an ectopic locus (*Ptn*) does not increase RDC whereas it does at another one (*Cttna2*) with similar control of transcription activity. Again, with this result, the authors' data strongly suggest that RDC occurrence cannot be explained in a general manner by transcription, nor is it by the presence of R-loops (as mentioned in page 10 and Supp Table S1). I have the impression that the authors describe a series of single-locus events/properties but are unable, at this stage, to draw or to identify general features for RDC formation. Finally, I also underscore that the title of their manuscript is: "Transcription-Replication Conflicts shapes DNA break dynamics". If the authors want to demonstrate a direct link between transcription and DNA breaks occurrence, they should provide evidence that this is a generally common feature of RDC, which is not the case at the moment.

- Throughout the manuscript, I find that some informations are hard to find. For example, in Fig 4C, we know that 85 RDC were taken into account, but how many fell in the HO vs CD clusters? Are they similar in size? Another example is the absence of scale when authors show DSB density in Figures 2 and 3, which prevents a reader to compare the various loci in terms of DSB occurrence.

Minor points :

1. Introduction, page 3, line 20. Authors are comparing Recurrent DNA break clusters and Common Fragile Sites and state that they differ in terms of DNA replication. It would be informative to recapitulate in one sentence the characteristics of CFS to fully appreciate in which aspects they differ from RDC.

2. Results, page 6, lines 5-11. I'm not sure whether the authors also included DSBs emanating from regions that are different of the CRISPR-Cas9 site in their dataset. Could the authors explain the

rationale and state clearly if they include or not other chromosomal regions than the CRISPR-Cas9 cleavage site.

3. Figure 2 B,D, F and H. DSB density in these figures seems to be associated with some sort of peak calling or a thresholding methodology as we can see a dash line on graphs yet I did not find a description of these dashlines in the figure legends, nor I found how it was determined in the text. This is quite an important point since the authors want to claim that forks directionality determine DSB orientation. It is even more important given the fact that for example we can clearly see: (1) signals for Dtel in the example of the Large gene below the dash line (Fig 2D) and (2) a Dcen peak in Sdk1 (Fig 2H) that is marked with a star even if the peak stays below the dashed line. Additionally, the authors mention that not all the regions analyzed behave similarly in a given context (e.g unidirectional: 20/35 exhibit a single peak DSB signal). To facilitate data visualization and interpretation by readers, I advise the authors to quantify the enrichment for both Dcen and Dtel in these different contexts and show the results in a graph where it is possible to see individual regions (e.g violin plots or else) in addition to the already represented data which are graphical and seems to use only the thresholding effect. I also found intriguing the absence of scale on the DSB density charts. This is of paramount importance to allow readers to compare the frequency of DSB in the different loci shown.

4. Fig2. The frequency of DSB following the expectation is indicated for unidirectional (20/35) and biphasic replication (5/9) but not for inward and outward moving forks. Does it mean that all regions analyzed in both contexts behaved similarly ?

5. Figure 3, here the authors show DSB density in the presence of Aphidicolin +/- ATR inhibition. It would have been interesting to show, on the same figure at the same scale, the DSB density in cells without treatment with Aphidicolin in order to estimate if there is already an increase with Aphidicolin treatment alone. This is even more relevant since the authors quantify DSB amount with various doses of aphidicolin in a subsequent panel (Fig 3E).

6. Figure 3 and Results page 8. The authors conclude in a very general way that they "demonstrated that genomic undergoing early replication in S the phase, as opposed to regions replicating during the median or late phases, display dormant origin activation upon ATR inhibition". This is a bold statement considering that the authors extrapolate this conclusion from the analysis of a single early replicated locus and a single mid replicated locus.

7. Figure 3, we can't find information relative to the duration of treatments (APH and VE-821) nor if the different doses of Aphidicolin employed activate similarly the S phase checkpoint kinase in NPCs.

8. Figure 4, it would be informative to depict movement of the transcription machinery in the same way than the replication machinery in the various panels.

9. Page 9, Results, consider reformulating the sentence at lines 15-16.

10. Page 9, Results, line 20. I don't understand why the authors refer to "The ideal scenario" ?

11. Figure 5 is lacking a legend for the colors used in panels C and D.

12. Figure 5, the authors need to be explicit, in the results description, if they are referring to samples treated or not with APH (e.g page 11, lines 6-7).

13. Figure 6: The authors found that the short isoform transcription of Nr1h3 is not leading to RDC formation, contrary to the long-isoform. Again, we can wonder why. Is it due to the fact DNA replication never reaches the transcription site of the short isoform ? What distinguishes the short vs the long isoform in terms of transcription ?

14. Figure 7: the comparison between the engineered Ptn locus and Ctnna2 is not clear. Was Ctnna2 also engineered in the same way and put under the control of Dox responsive element ?

Overarching Response to the Reviewers

The authors want to thank all the reviewers for their constructive suggestions. This manuscript
has undergone a significant revision based on the new data presented in the revised version.

We would like to start responding to all reviewers with a core issue. All reviewers pointed out
that the data displayed in the current manuscript were not sufficient or clear-cut to support our
conclusions. In particular, the RDC behavior at the early/out-moving forks does not quite fit our
proposed model. We fully acknowledge the problem presented in the original manuscript.

In the original manuscript, we defined the fork direction using the published high-resolution
Repli-seq data generated from wild-type neural progenitor cells not treated with aphidicolin.
*Xrcc4/p53*-deficiency or aphidicolin treatment may change the replication start and end positions.
Specifically, Sarni et al. and Brison et al. suggested that DNA replication timing was advanced in the
aphidicolin-treated cells at the boundary of common fragile sites than in the controls. Advancing
replication initiation may create new initiation zones to be computationally identified. The new
initiation zone may influence fork direction determination, which is critical for our manuscript.
Besides, reviewer 3 questioned whether DNA replication reaches the RDC loci under replication
stress. To collectively address these questions, we performed the 16-fraction, high-resolution
replication sequencing on untreated or aphidicolin-treated *Xrcc4/p53*-deficient neural progenitor cells
(revised Fig. 1). In the revised manuscript, we used the Repli-seq data from aphidicolin-treated NPC
to predict replication direction. We also used this opportunity to analyze the replication features to
narrow down genomic regions that can be reliably assigned for fork direction.

[redacted]

Figure R1. Comparison of Newly Generated and Published High-Resolution Repli-seq Datasets. (A) Normalized high-resolution Repli-seq heatmaps for chr1:30,000,000–60,000,000 in the F121-9/CAST NPCs (signal in red, Zhao et al., 2020) and in the *Xrcc4/p53*-deficient NPCs (signal in black, this article). S phase fractions are shown at the left-hand side. **(B)** The ring chart showing the percentage of features observed in high-resolution Repli-Seq heatmap.

We first compared our untreated high-resolution repli-seq datasets to the neural progenitor cell
data from Zhao et al. with the published analytic toolbox (Zhao et al. 2020). Figure R1A shows the 16-
fraction data from Zhao et al. (signal colored in red) and from us (in black) for mouse chromosome 1.
I hope the reviewers can appreciate that new and advanced initiation zones are present in Xrcc4/p53-
deficient neural progenitor cells despite the high similarity between the two datasets. The majority of
the mouse NPC genome consists of timing transition regions (TTRs) and constant timing regions
(CTRs) (Figure R1B). In the *Xrcc4/p53*-deficient NPCs, 16 – 20% of the genome contains initiation
zones (IZ), and about 1% of the genomes contain termination zones smaller than 100 kb (Figure R1B).
We concluded that the high-resolution Repli-seq generated in *Xrcc4/p53*-deficient NPC preserves
replication features at the exact resolution as in Zhao et al. datasets. We described the high-resolution

Repli-seq experiments and the analyses in the revised manuscript on page 4 and summarized in Table
 S4. There are slight differences between the proportion of IZ called in our datasets. We have no
 intention to compare the subtle differences between *Xrcc4/p53*-deficient NPC and wild-type F121-
 9/CAST NPCs.

 Next, we applied our fork direction prediction algorithm (described in the original manuscript,
 Figure 1) to the high-resolution Repli-seq dataset from untreated and aphidicolin-treated neural
 progenitor cells. The results are described in the revised manuscript under “Replication Direction
 Maps for XRCC4/p53-deficient Neural Progenitor Cells” in the Result section (page 4). It is worth
 mentioning that replication sequencing data from Zhao et al. did not contain signals for chromosome
 X. Our new datasets resolved this issue.

 Surprisingly, we found that a substantial proportion of “outward-moving,” “unidirectional,”
 and “complex” RDCs became “inward-moving” under aphidicolin treatment. The replication direction
 defined using our APH-treated high-resolution Repli-seq data is denoted in the revised Table S5. In
 addition, we also characterize the broad IZ zone and broad CTR, which are rapidly replicating regions
 with multiple forks within a 50 kb bin. Within these regions, one cannot assign fork direction. We
 avoid analyzing RDCs present in these areas regarding fork directions. By the combination of fork
 direction and broad IZ/CTR feature, we defined 87 “inward-moving” (with two slopes going inward,
 sometimes with short CRT), 15 “unidirectional” (does not contain IZ, TZ, or CRT), six “outward-
 moving” (contain one IZ and two slopes going outward), 12 “complex” (which has IZ and TZ features
 within the same RDC), and 32 “undefined” RDC. Significantly, **most RDCs are DNA breaks at the
 TTR, where sparse replication origins connect unidirectional forks (Figure R2).** These
 observations are described in the revised manuscript under the Result section on pages 5-8.

Figure R2: The *Npas3* RDC consists of orientated DSBs at TTR; revised Figure 2B,C. (A) The figure illustrates single-ended DNA breaks at rightward- (left) and leftward-moving forks (right). The light blue DSB end at the rightward-moving fork is linked with centromeres, maintaining its centromeric orientation (Dcen) when joined with the “bait” DSB end (green). Conversely, pink DSB ends at the leftward-moving forks are linked with telomeric sequences, preserving their telomeric orientation (Dtel) upon joining with the “bait” DSB end. (B) Top: A heatmap displays high-resolution Repli-seq data in aphidicolin-treated NPC. A blue vertical line denotes the predicted termination zone, while two red dashed lines indicate the genomic positions with the earliest timepoint within the two Initiation Zone (IZ). Colored arrows annotate the replication direction. Middle: A smoothed histogram depicts the density of DNA breaks at the recurrent DNA break cluster (RDC) and its surrounding area, with a plotted window size of 3 Mb. The Y-axis represents the extended interchromosomal translocation within a 25-kb kernel. The density of inter-chromosomal translocated junctions at the centromeric end (Dcen; blue) and the telomeric end (Dtel; pink) is illustrated.

 The second core issue concerns the quality and quantity of data presentation. To enhance the
 appreciation of the symmetry between RDC break density and replication fork directions, we
 compiled, for each RDC, the RDC break density and high-resolution Repli-seq data in the same graph
 (Figures 2, 3, S3, and S4 in the revised manuscript). **Figure R2** is an example of RDC at the *Npas3*
 locus. The centromeric and telomeric DNA break ends (Dcen and Dtel) were shown as blue or pink
 peaks. The Repli-seq signal from fraction S1 to S16 at the same genomic area is shown below. The
 termination zone and nearest initiation zones at or around per RDC were shown as blue or dashed red
 lines, respectively. Gene bodies in the plotted area were annotated in light (centromeric to telomeric)

or dark green (telomeric to centromeric). Lastly, the range of the entire RDC and the Dcen and Dtel
peak range were annotated. We presented most “inward-moving”, “unidirectional”, “complex”,
“outward-moving”, and “undefined” RDCs. A summary for all classes is shown in revised Figure 3E.

As reviewer 1 suggested, we also performed strand-specific DNA:RNA hybrid analysis
(DRIPc-seq). Because of the newly generated high-resolution repli-seq and the strand-specific
DNA:RNA hybrid data, we extended our analyses and asked if DNA break density correlates to the
density of co-transcriptional DNA:RNA hybrids. In summary, we found that **the density of transient**
**co-transcriptional DNA:RNA hybrids positively correlated with DNA break density**. These new
data are presented in revised Figures 5 and 6.

We also conducted pairwise Repli-seq experiments for ATRi and APH-treated NPCs to
complement the results presented in the original Figure 3. Unfortunately, the experiment failed. We
recovered very few BrdU labeled DNA, presumably due to the insufficient BrdU incorporation (45
minutes) under this condition. As we cannot directly compare DNA replication status and DNA break
density under the ATRi+APH condition, we withdrew the ATRi+APH experiments (original Figure 3)
from the revised manuscript.

We have addressed all the concerns and believe the manuscript has improved significantly. We
have rewritten most of the manuscript. To assist reviewers in identifying the original text, **we colored**
**the unchanged text in blue in the revised manuscript**. We believe the new datasets are essential to
transform this manuscript. Below, please find our point-to-point response to reviewers’ comments.

24 **References**

- 1. Sarni et al. <https://doi.org/10.1038/s41467-020-17448-2>
2. Brison et al. <https://doi.org/10.1038/s41467-019-13674-5>

**Reviewer #1 (Remarks to the Author):**

The authors of Ionasz and Corazzi et al used Repli-seq and OK-seq datasets to derive by
neural network training, replication termination zones that would link with replication initiation
zones to correlate and explain the formation of chromosome orientation-specific solitary
DSBs in previously identified transcriptionally active recurrent break cluster (RDC) genes of
ES cell derived neural progenitor cells deficient in Xrcc4 and p53.

The authors then designate each RDC 1 of 5 categories and highlight their most
distinguishing features. Using DRIP-seq, they demonstrate that the replication-dependent
RDCs formed do not appear to be affected by paused transcription and subsequent transient
RNA:DNA hybridization that would constitute an R-Loop.

Notably, great effort was put forth to demonstrate differential effects of transcription on the
formation of RDCs, which varied in significance by locus. Thus, the authors identified multiple
factors driving orientation-specific DSB formation that include replication fork direction,
replication timing, and locus-specific effects of transcription on RDC activity, which for the
latter two parts speculatively implicates local chromatin differences according to cell type as
a contributing factor to the overall RDC generation model.

Overall the work provides compelling evidence that TRCs are causal to the formation of
many RDCs but that the slowed replication fork DSB generation is affected by more than just
head-on transcription for some RDCs.

**Response:**

We thank reviewer 1's comments on our compelling evidence that TRCs are causal to
forming the majority RDC.

**Major points**

Point 1: Although the DSB directionality matches well with replication directionality as
described, there are two instances where statements made suggest either an RDC category
behaves in a common fashion for like RDCs in that category, rather than just one RDC
displaying this phenotype from the data shown, or an alternative interpretation of the data
shown could be made.

**Response:**

We agree with reviewer 1's comments that more data should be shown to support our
observation. As mentioned in "Overarching Response to the Reviewers", we now provide
DNA break density, transcription direction, replication sequencing, and fork direction for all
RDCs in the revised manuscript under Figures 2, 3, S3, and S4.

Point 2: First, as described in the abstract "This pattern, however, reverses in early-
replicating DNA" and on page 4 lines 8-9: "We found that the orientation of DSB ends
outward moving fork directions were opposite the proposed model", it is clear from Fig2 F
one example shown looks to have red/blue twinning rather than the blue/red twinning seen
for inward fork DSBs but the 2nd example, March 1, does not have the red/blue twinning and
looks more similar to the inward pattern

**Response:**

According to the aphidicolin-treated 16-fraction Repli-seq data, the locus *March1* represents
an “undefined” RDC due to the lack of significant initiation features, and fork directions
cannot be determined (Fig. S4). Yet, the Repli-seq suggests inward-moving forks may
progress into the *March1* locus (Figure R3).

Figure R3: DNA replication and DNA break alignment at the *March1* locus; a panel in the revised Figure S4. Top: A heatmap displays high-resolution Repli-seq data in aphidicolin-treated NPC around the *March1* locus. Figure is organized as described in R2.

Point 3: Can the authors either firm up this opposite DSB pattern seen for the outward
moving category with other RDCs? Otherwise the statements made really only apply to one
locus of that RDC category and become less significant.

**Response:**

According to the new 16-fraction Repli-seq datasets, overall outward-moving RDCs were
reduced from twelve to six (Figure S4. of the revised manuscript and Table S5). We carefully
examined the remaining “outward-moving” RDCs and found the distributions of *Dcen* and
*Dtel* at RDCs exhibited overlapping patterns in four of them (Fig. S4). In addition, only *Tiam2*
RDC preserves the *Dcen* and *Dtel* features we proposed in the original manuscript. With the
low number of “outward-moving” RDCs and no common feature among them, we withdrew
the originally proposed model for “outward-moving” RDC.

Point 4: Second, the Large gene RDC is listed as unidirectional (rightward) in Fig 2D based
on RDC designation, but data in Fig 3, which look reproducible to Fig 2 data, show a pink
late replicating leftward peak for Large (Fig 3C) just outside of the early replicating RDC
designated area that increases with ATRi; clearly this does not connect RDC seeds
according to described calling parameters but this could also be categorized as an inward
fork DSB and thus implies more than one category may satisfy an RDC or that the RDC
calling for this locus is not accurately depicting the translocation data presented.
Furthermore, to support the point made for early replication sites firing dormant origins with
ATRi in Fig 3B, it could be argued that the shoulder of the blue peak in Fig 3C is shifted in a
similar fashion. This was not noted in the manuscript and may need to be described more
fully in the results.

**Response:**

We acknowledge reviewer 1’s comment on the Large 1 locus. This question could be linked
to the significance cutoff of the RDC calling algorithm. RDC and the *Dcen*/*Dtel* islands were
defined by MACS2, a bioinformatic method that compares local signal versus background
with a significance cut-off. An island with a q value greater than 0.1 will not be considered
significant. In the original manuscript, the pink DSB density at the *Large* gene locus was

below the threshold. In addition, the Dcen and Dtel islands were called independently of the
RDC island. In the revised manuscript, we annotated the significant Dcen, Dtel, and RDC
island under the “Annotation” box under each multiomics plot in Figures 2, 3, S3, and S4.

We also revised the RDC calling under the Method section on page 22, lines 6-18, quote:
:

“For peak calling, we extended LAM-HTGTS junctions by 50Kb symmetrically in both
directions, and pileup islands were determined for telomeric-only (Dtel), centromeric-only
(Dcen), and all junction orientations (Dtel + Dcen). A negative binomial model for estimating
the expected pileup value for each chromosome/condition/junction-orientation triplet was
derived, and a p-value was calculated for each pileup value concerning model expectation.
Regions with a p-value below 0.01 joined (maximal gap 10Kb) to create seeds. These seeds
were further joined with other seeds (maximal gap 100Kb) to form islands. Islands are
extended up and downstream to include regions below 0.1 significance. Overlapping
orientation-specific islands are further joined to form an initial RDC list that is further filtered
to contain at least 100Kb below 0.01 p-value and be of at least 300Kb in length when
considering extended regions. The broadest range of all overlapping and significant islands
determined RDC. The RDC-calling algorithm is deposited on GitHub under the link below
(https://github.com/brainbreaks/DSB_Paper).”

Point 5: Given the value in learning about which orientation DSBs are formed at TRCs, the
authors should discuss further whether the ends presented are from the leading or lagging
strand or if it is not clear. Related to this point, figure 2 diagrams were helpful but seemingly
showed both strands of each fork as broken, which would not be solitary.

**Response:**

We thank the reviewer for pointing out the critical point. Our experiment setting cannot
assess whether DSB occurs at the leading of the lagging strand. Since we cannot prove
whether RDC break ends are enriched at leading or lagging strand, we chose not to specify
the strandness but rather show one break per fork instead. We have corrected the revised
manuscript's diagram in Figures 2 and 3.

Point 6: Although the first half of the manuscript was devoted to categorizing RDC subsets
based on replication fork progression, it was not clear whether replication fork progression
status in the RDC (i.e., DSB position relative to initiation/termination) itself plays a greater
role in determining the extent to which transcription is a significant causal mechanism. For
instance *Ctnna2* and *Foxp1* had opposing results despite both being inward moving (late
Rep) DSBs. *Nrxn1* is listed as complex in Table S1, mid-S phase in Fig S6 and *Ptn* goes
from late in ES cells to mid-early in NPCs which should translate to unidirectional/outward for
the latter. Connecting these points, perhaps, somewhere in the last 2 paragraphs of the
discussion might be useful to make the manuscript more cohesive overall.

**Response:**

We apologize for the confusion regarding the *Foxp1* locus in the original manuscript. The
promoter and enhancer of *Foxp1* (Fig. 5D of the original manuscript) were not removed. This
locus was shown as a control that RDC can be induced in the *Ctnna2*-ape neural progenitor
cell lines. This issue was also mentioned by reviewers 2 and 3, as these experiments do not
add value or explain overall RDC behavior. In the revised manuscript, we have enhanced the
understanding of the genome-wide linear interaction between transcription and DNA
replication. The findings from the single locus experiments have become less significant and

do not contribute to the overarching picture of the current manuscript. Therefore, we
excluded the original Figures 5, 6, and 7 from the revised manuscript.

Point 7: Can the authors speculate on what twin versus overlapping peaks may imply with
regard to TRCs of inward or outward moving forks?

Response: In the revised manuscript, we demonstrated that most twin peaks appear at two
separate TTR (Fig. 2, S3, 3, S4), whereas overlapping peaks appear at the R-loop persist
region or the broad initiation zones. We concluded that the twin peaks are derived from DNA
breaks at two TTRs. We speculate that the overlapping pattern at the R-loop enriched area
might be due to the position of DNA breaks ahead of the fork but not at the fork. The model is
proposed in Figure 7 in the revised manuscript and here (Figure R4). Nevertheless, we
cannot exclude the possibility that the density of active forks is higher than one per 50 kb,
which is below the resolution of our assays.

We incorporated the following texts in the revised manuscript on page 9, line 18 to page 10,
line 5:

“... Among the 152 RDCs analyzed, two-thirds of them did not contain R-loops. We found
one-third of RDCs contained one to nine R-loops, and only four RDCs (Ash1l, Kihl29, Sil1,
Prkcz) harbored more than ten R-loops (Table S5). In the “outward-moving” Kihl29 RDC,
Dcen and Dtel did not align with the replication fork directions but to the R-loop position.
Similarly, the overall DNA break density aligned with the R-loops for the Sil1, Ash1l, and
Prkcz RDCs (Fig. 5A). The fork directions could not be determined for Ash1l and Prkcz loci
as they were present at the broad initiation zones (Fig. S4). This observation suggests that
R-loop persistence alters the proportion of Dcen and Dtel, leading to RDCs displaying
“overlapping” peaks. In total, the Dcen and Dtel peaks significantly overlapped in 30 RDCs,
22 of which presented at broad initiation zones and contained persisting R-loops (RDC in
Dst, Kihl29, Trappc9, Prkcz, Tmem132b, Peak1, Plekhg1, RDC-chr9-35.4, Msi2, Slc39a11,
Cdkal1, Zmiz1, Samd5, Cdkal1, Samd5, Cep112, Csmd2, Rere, Ptn, Ash1l, Tln2, and
Gm12610; Table S5) “

And under discussion, on page 14, lines 15-24, quote:

“For “outward-moving” and RDCs within broad initiation zones, we observed that the DNA
break positions are in substantial accordance with the presence of the R-loop (Fig. 5A). We
speculate these RDCs share the pathway that creates ERFS. Multiple replication origins are
proposed to be simultaneously fired within the initiation zones, leading to “active” DNA
replication. At this region, active and frequent origin firing may collide with the R-loop, leading
to DNA breaks ahead of the fork. These processes may generate double-ended DNA breaks
that are not solitary (Fig. 7). Mechanisms for DSB ahead of the fork were previously
proposed by investigating the rDNA genomic in yeast. Nevertheless, we cannot exclude the
possibility that the density of active forks at CTR is higher than one per 50 kb, which is below
the resolution of our assays.”

Furthermore, we have observed that there is a partial overlap between Dcen and Dtel at the
broad CTR where R-loops are absent. This observation made us speculate that these breaks
occur at late-firing replication forks spaced less than 50 kb apart. We discussed this
possibility and their relationship to CFS under the Discussion on page 14, lines 2-14:

“Twenty-three “inward-moving” and 12 “undefined” RDCs are present at the broad late CTRs
(Table S5). As the DNA break density increment yet represents a dosage-dependent effect in
cells treated with aphidicolin (Fig. 4A, Prkg1), we believe these are also DNA breaks
resulting from replication stress. Intriguingly, as proposed previously, CTRs are genomic

regions where the replication origins are only fired at the late S phase. In RDC containing
 broad late CTR, Dcen and Dtel density overlap with the CTR (Fig. 2E, G, and S3),
 suggesting that these DNA breaks primarily occurred at the last S phase fractions. In
 addition, the high-resolution Repli-seq data indicated that DNA replication is completed at
 most CTR regions (Figs. 2, 3, S3, and S4) with a few exceptions at the genomics sequences
 underlying Magi1, Ccser1, and Grid2 RDCs, where a gap in the CTR was observed (Fig. S3
 and S4). This gap is likely due to underreplication at the center of specific RDC-containing
 genomes. Hence, a subset of broad late CTR-containing RDC may share the DSB-initiation
 mechanism as CFS(Fig. 7). ”

Figure R4. RDC DNA Breaks Orientation Dynamics; revised Figure 7. Figure depicts the DNA break position relative to the replication fork at RDCs within the TTR region (top), broad and late CTR (bottom-left), and broad initiation zone (bottom-right). Replication fork direction and DNA break orientation are shown in Fig. R2.

 **Point 8:** The reader may be better prepared to understand outward vs inward moving
 contexts by explicitly connecting them to early replication and late replication; logic would
 then indicate unidirectional would then be mid-S phase replication. However, if there are
 examples of inward/early or the opposite, then those conditions should be discussed as then
 you would have converging replication in early replicating genes.

 **Response:**

 Most RDCs are composed of timing transition regions (TTRs). As mentioned, TTRs are
 regions with fewer replication origins than initiation or constant timing zones; thus, one
 cannot state a timing. We emphasized this point in the revised manuscript. The replication
 profile for all RDC-containing gene loci is shown in the revised manuscript for the reader to
 inspect the replication timing.

 The classical definition of "early-replicating" genes applies to the broad IZ zones where the
 direction of replication forks cannot be determined. We found six "inward-moving" RDC loci
 (Sil1, Col4a2, Qk, Zmiz1, Rere, and Msi2) where the genomic sequences underneath were
 replicated within the earlier S phase fractions (Fig. S3). RDCs in these regions lost the "twin-

peak" signatures. For instance, Dcen and Dtel largely overlap at Qk and Rere loci. This
finding suggests that early-replicating genomic regions follow separate TRC mechanisms
uncoupled from the fork progressing direction. We have included the above descriptions in
the result section, between page 7, lines 1-5.

Point 9: Table S1 should include some type of NPC replication timing designation for each
RDC especially if there are exceptions to the above comment.

**Response:**

As explained earlier, TTR cannot be given a replication timing; we opt not to define RDC
according to time. Alternatively, we created another column to include the replication features
in the revised Table S5.

Point 10: The DRIP-seq analysis describes RDCs as having minimal R-loop accumulation,
consistent with an earlier publication, and in RDC genes that are responsive to forming DSBs
from transcription alterations, suggesting that replication-driven RDC effects are not
necessarily due to transcriptional pausing that would promote DNA:RNA hybridization.
However, no mention was made to describe any positive R-loop peaks from early replicating
genes to demonstrate some level of rigor for a negative result.

**Response:**

We thank the reviewer for pointing out the issue. The data concerning positive R-loop peaks
are now shown in the revised Figure 5A and here (Figure R5). The results were described on
pages 9-10 under the "RDC Displays Differential Accordance to DNA:RNA Hybrids" section
and in the earlier response to point 7 raised by reviewer 1.

Point 11: Furthermore, speculation is presented in the discussion for even more transient or
different (dual strand) DNA:RNA hybrids contributing to the mechanism despite the negative
correlation of DRIP-seq peaks and RDCs. It would be helpful if the authors can clarify this
discrepancy in terms of R-Loop contribution to RDCs. Perhaps the DRIP-seq findings could
be complemented with strand specificity of DRIP-seq peaks using the same protocol
publication (DRIPc-seq) that is cited in the methods or use nuclear DRIP-seq to potentially
support the conclusion and discussion point.

**Response:**

We thank the reviewer's suggestion. We performed the DRIPc-seq in *Xrcc4/p53*-deficient
ESC-NPCs, and the complete results were described in Pages 10-11 and Figure 5 in the
revised manuscript. The new DRIPc-seq datasets allowed us to analyze DNA break density

with multiple dimensions. These results are shown in the response to reviewers below.

First, the new DRIPc-seq data supports the original DRIP-seq experiments on the R-loop
position. We found 7336 R-loops that DRIP-seq defined contain significant template-strand-
specific DRIPc-seq peaks. These data are shown in the revised manuscript. Second, we
defined co-transcriptional DNA:RNA hybrids present in the RDC-containing gene at the
coding strand (Figure R6 and revised Figure 5), which was not detected as R-loops using the
DRIP-seq protocol.

Second, we found dual-strand DNA:RNA hybrids are significantly enriched in RDC compared
to actively transcribed, non-RDC-containing long genes. These data are presented in revised
Figure 5 and here (Figure R7). The reviewer mentioned that the dual strand DNA:RNA hybrid
could be co-transcriptional transient antisense RNA resulting from RNA polymerase II stalling
(Eva Petermann, Li Lan & 2022 NRCMB). We intended to compare the dual-strand
DNA:RNA hybrids with the antisense transcription activity with the GRO-seq datasets.
Unfortunately, we did find significant antisense transcription in these loci with our GRO-seq
datasets, presumably due to the low transcription activities overall at RDC-containing genes.

The dual-strand DNA:RNA hybrids may also represent the Okazaki fragments generated by
PrimPol when the leading strand DNA polymerase is stalled. However, in this case, the
hybrid should be present on one of the strands, not both. We excluded the possibility that
dual-strand DNA:RNA hybrids are the PrimPol-mediated Okazaki fragments.

Figure R7. Dual strand DNA:RNA hybrids density is higher in RDC-containing genes than other long and actively transcribed genes in NPC; revised Figure 5 O. The density of dual-strand DRIPc-seq peaks in RDC-containing genes versus in genes longer than 100 kb without RDC. The Mann-Whitney test determined statistical significance. ****P <0.0001, while n.s. denotes not significant.

Point 12: There is a materials section detailing the various sequencing methods used but this reviewer could not find data deposition details for GRO-seq (both ES and NPCs) and DRIP-seq in the manuscript but found one GRO-seq (ES Cast.129 cells) and DRIP-seq in the two indicated GSEs. NPC GRO-seq, if done previously, should be referenced with the GSE number.

Response: We generated GRO-seq datasets on our own for this manuscript. The specific data deposition is now indicated in the revised manuscript. Specifically, GRO-seq libraries for Xrcc4/p53-deficient NPC were deposited under GSE233842. LAM-HTGTS libraries were deposited in GSE233842. DRIP-seq and DRIPc-seq libraries were deposited in a new GEO session, GSE254765, together with the high-resolution Repli-seq libraries. The authors realized that the naming system for the GSE233842 datasets was very confusing. We have updated the sample names so that the experiment type, repeat number, and conditions are displayed.

Point 13: While it is appreciated which newly generated baits were included for LAM-HTGTS, some quantitative value (#s or %) of what is newly added relative to what was combined with previous studies or which figures would be completely new would help benefit readership understanding of what is added and clarify how panels covering the same loci in prior publications are different.

Response:

The authors understood the reviewer would like us to clarify which RDCs are new. We included the following texts in the revised manuscript on page 4, lines 15-24, quote:

“The RDC collection (Table S2) described in this article is characterized by combining the published^{22,23} and newly generated datasets. A detailed description of the RDC calling process can be found in the Methods section. We characterized 152 RDCs, 78 described previously (Table S2). The newly characterized RDCs are all in genomic regions containing actively transcribed genes (Table S2). Consistently with the findings of previous RDC studies, genes underlying the newly identified 74 RDC show an overrepresentation of neuronal functions and encode proteins controlling cell adhesion and synaptic functions (Table S3). In this article, we analyzed the relationship between DNA breaks and the linear interaction of genomes under the 152 RDCs.”

We would like to emphasize that it is more a sequencing-depth issue than the number
of libraries used in our analyses. The new datasets (23 APH-treated and 19 untreated LAM-
HTGTS libraries) generated from this article were sequenced under NextSeq 550. NextSeq
produces five times more reads per library than its predecessor, Miseq, which was used to
produce LAM-HTGTS libraries for the published datasets (59 APH-treated and 59 untreated
LAM-HTGTS libraries by Wei et al., 2016 and 2018). We noted this difference under the
“LAM-HTGTS Libraries used in this article” paragraph in the Method section in the
supplementary information.

For the Corrazi and Ionasz manuscript, we also designed the RDC calling algorithm
to consider whether Dcen or Dtel islands remain significant within RDC. This calling
approach differs from Wei 2016 and 2018, in which all translocations, regardless of
orientation, were included in the calculation.

When applying the new algorithm to the published LAM-HTGTS datasets (Wei et al.
2016 and 2018), we only validated 28 RDCs displayed significant Dcen and Dtel islands. All
28 RDCs were previously described as RDCs. When applying our algorithm to the datasets
generated by Corazzi and Ionasz et al., we identified 143 RDCs.

We believe the discrepancy is due to the size of the individual library. The “Wei”
libraries are smaller (~ 10K). The library size from Corazzi and Ionasz is much larger; on
average, we recovered ~ 30,000 junctions per library. As our peak calling algorithm also
considers reproducibility across experiments – meaning the peak has to appear in three
independent libraries – we lost many RDCs from the published dataset due to lower
sequencing coverage.

The following text was added to the Methods section, under RDC calling between Page 22,
lines 18-27:

“The algorithm used in this manuscript aimed to define orientation-specific islands and join
islands to form RDC. The algorithm called 28 RDCs from previously published datasets ^{22,23},
and 143 RDCs from the LAM-HTGTS datasets generated in this manuscript. All 28 RDCs
called using the orientation-specific algorithm are previously defined RDCs. Due to the
smaller library sizes (~10k per experiment) in the previously published datasets, the newly
generated libraries (~30k per experiment) contributed to most of the RDC analyzed in this
manuscript. We annotated whether the RDC is newly identified or described previously in a
column in Table S2. The additional RDC identified by this combinatorial approach is
contributed by enhanced data depth, as the new RDCs already display slightly enhanced
DNA break density in the previously published datasets.”

Point 14: Page 8 lines 11-13: Provide a reference or additional RDCs indicating the early but
not mid/late replicating sites fire dormant origins with ATRi. This could be added to Table S1
for RDCs with enough power.

**Response:**

Menolfi et al. (DOI: 10.1038/s41467-023-39332-5, 2023 Nat Comm) suggested ATR tempers
the pace of origin firing at the early S phase in unstressed cells. As we removed the ATRi
experiments from this manuscript, we did not add this reference in the revised manuscript.

Point 15: Page 15 line 2: There is no Figure S5E in this version of the manuscript and either
this data should be included, given that the reference provides evidence that contrasts with
prior observations seen in DT40 and human cancer cells, or the discussion point should be
removed.

Response:

We apologize for this mistake. The figure was presented in Figure S4D in the original manuscript. This figure has now been removed from the revised manuscript.

Minor points

Generally, please read through carefully for grammatical errors and incorrect figure panel citations.

Page 2 line 5 became → becomes
Page 5 line 20 viewpoint → bait viewpoint

Response: We have corrected these mistakes.

Page 5 lines 24-27 needs more clarification and might be better described as it relates to within each RDC rather than the genome as translocations to sites outside of RDCs may have other mechanisms in play.

Response: We thank the reviewer's suggestion. We included the following text on page 6, lines 6-9, quote:

"It is important to note that mechanisms other than RDC may be involved in translocations to sites outside of RDC, such as off-target sites generated experimentally with CRISPR/Cas or recombining immunoglobulin gene loci, which have been described elsewhere ^{28,31,32}."

Page 8 line 11: Capitalize new sentence

Response: This section has now been removed from the revised manuscript.

Page 8 line 18: DTM is not clear, per thousand interchr. DSBs, bps, or reads?

Response: when referring to DNA break density, the unit is "DSBs per ten thousand interchromosomal translocations" in the revised manuscript.

Page 10 line 11 R-loop presence was absent → R-loops were not detected

Response: This sentence has been replaced by new texts in the revised manuscript.

Page 14 line 6 MUS81 are → is
Page 23 lines 20, 23, 28: it appears the wrong figures are being referenced.
Page 24 line 1: Fig3C is not the correct figure reference

Response: These contents are not present in the revised manuscripts.

Fig. 2B legend: not clear what the units of DSB density are: DSBs/100kb as an absolute number or percentage or some type of 100kb sliding window? Absolute RDC numbers are not reported so it is difficult to evaluate which RDC patterns are more significant.

Response: We added a Y-axis to each omics figure in the revised manuscript. In the case of RDC density presented in Figures 2, 3, S3, and S4, the Y axes represent extended junction counts within one 25 kb kernel. We added the non-extended junction number per RDC at Dcen or Dtel orientation in the revised Table S5.

Fig. 4C legend should indicate how many RDCs were used as was indicated for Fig 3E; also the 4C panel is difficult to view the stripes with the color scheme shown as they both look like solid colors at low paper viewing magnification; the panel legend references two conditions but the graph has three colors (DMSO). This is not clear but an otherwise excellent panel from a scientific viewpoint.

Response: We thank the reviewer’s appreciation of the graph. We analyzed 87 “inward-moving” RDCs, 15 “unidirectional” RDCs, and 12 “complex” RDCs in the revised Figures 4 and 6.

Figures 5C,D and 6C,D should have some pictorial indication in the panel as to which are aph treated.

Fig. 6 legend (D) describes a figure 4D panel that does not exist in this version.

Fig. 7A not clear what the extra red circle cross is indicating below mPGK

Fig 7B,C,E,F should indicate the cell line and genotype used since this is different from other main figures that were NPC; *Xrcc4^{-/-}p53^{-/-}*.

Fig 7C library numbers should also be reported in the legend to be in alignment with Fig 7E,F

Fig. S5B is missing description of promoter colors

Response: These figures are no longer present in the revised manuscript.

Reviewer #2 (Remarks to the Author):

Point 1: The Ionasz and colleagues are studying a very interesting and fundamental
phenomenon of replication-transcription collisions in mammalian cells. I generally find the
data they generated very interesting and studying directly dose-dependence of eg aphidicolin
treatment or RTC is very interesting and deepens our understanding of mechanisms of RTC,
as well as studies of RTC at isoforms or the impact of R-loops.

Response:

We thank the reviewer for appreciating the interesting data.

Point 2: The paper have several weaknesses tough, one of the significant. The authors
consider that their main result and "ground breaking discovery" is that they are able to detect
1-ended DSBs in the sequencing data. However, detection of 1-DSBs in sequencing data
was reported several years ago and used to deducing local directions of replication forks
(Zhu et al. 2019, Fig. 5c, attached) (<https://doi.org/10.1038/s41467-019-10332-8>).

Response:

We are sorry that our original content resulted in this misunderstanding. The authors would
like to clarify that we have no intention to claim the novelty of the single-ended DNA feature
in the cells. To explain this point, we included the reference the reviewer suggested along
with Wills et al., 2017 under the sentence "DSB present at the fork as single-ended has been
shown before in yeast and mammalian cells." on page 6, lines 5-6 and referenced the papers
in the Discussion. To avoid confusion, we deleted the "groundbreaking" sentence.

Point 3: I am also very confused by the "model" that the directionality of the collapsed fork
will be reflected in the sequenced read and a sentence in the abstract saying that this
"model" is not correct in early replication.

Response:

We have corrected our model for the early-replicating RDC in the broad IZ zone. As most
early-replicating RDCs present in broad IZ zones, where one cannot determine fork
directions, we withdrew the previous model from the revised manuscript. This was described
in the response to reviewer 1, on page 5, lines 13-19, and summarized in Figure 3E.

Point 4: The relationship between fork direction and to which strand the resulting DSB would
map is rather straightforward and is explained in more detail in Zhu at al 2017
(<https://doi.org/10.1101/171439>) (Fig. attached). If the authors reach different conclusions it
can be caused by their predictions of local fork direction being inaccurate or by coexisting
fork directions and frequencies of one-ended DSBs not being a linear function of frequency
of forks traveling in different directions (since the authors and others before concluded that
heads-on collisions are more likely to result in DSBs).

Response:

We included the possibility that DSB may form ahead of the fork at the broad initiation zones
when the R-loop persists. We have corrected our model accordingly.

Point 5: "However, the exact orientation of DSB ends at the replication fork within living cells
remains an unanswered question." This statement is also not correct due to the reports in
Zhu at al 2019.

**Response:** This sentence has been removed from the revised manuscript.

**Point 6:** Another issue is that state of the art is not that well explained in abstract and
introduction. The information on the R-loops is very basic and not up to date. It is true that a
2016 paper reported seeing R-loops only in 2-15% of the transcribed genome, but with
technique improvement it has likely changed and is not a fundamental R-loop characteristic.
On the contrary, more recent findings on the R-loops could be added, for example emerging
understanding that many R-loops arise physiologically and do not promote RTC or DSBs,
unlike some pathological (or toxic) loops (Promonet et al. 2020
<https://doi.org/10.1038/s41467-020-17858-2>). More interesting fact that R-loops are typically
enriched at TSS and TTS but can also form inside the genes (e.g. Fig. 6 in Promonet et al.
(2020), <https://doi.org/10.1038/s41467-020-17858-2>). Moreover, Promonet et al. reported
that many R-loops are physiological and only subset of them is causing DSBs in a context
dependent manner.)

**Response:** We have referenced Promonet et al. 2020 in the introduction, on page 3, line 4.,
quote:

"..., while some R-loops arise physiologically and do not promote TRC or DSBs²⁰"

**Point 7:** Language of the paper is often confusing and would benefit from simplifying.

10-11 "These DSB ends possess inherent orientations, attaching themselves to either
centromeric or telomeric sequences on mammalian chromosomes"

I understand what authors are trying to say here, but talking about DSBs "attaching
themselves" to either centromeric or telomeric sequences creates confusing visual. Since
those DSBs result from broken forks, they simply inherit those forks' orientation.

**Response:** we have corrected the description in the Abstract:

"Leftward-moving forks generate telomere-connected DNA double-strand breaks (DSB) while
rightward-moving forks lead to centromere-connected DSBs." We also included new
illustrations in Figure 2A and Figure 7 to demonstrate telomere-connected DSB and
centromere-connected DSBs.

**Point 8:** "As replication stress intensifies, excessive DSBs at forks became sensitive for
rearrangements". How do we know which breaks are "excessive"?

**Response:** The word "excessive" is deleted from the revised manuscript.

**Point 9:** "Gaining an understanding of these orientations holds the potential to illuminate the
processes of genome rearrangements under conditions of replication stress." I understand it
boils down to knowing whether DSB originates from HO or collinear collision would help to
understand mechanism of genomic rearrangements? If so, an example would be helpful.

**Response:** we rewrote the sentence on page 3, lines 6-9:

"However, whether the orientation of TRCs matters in actual chromosomes exhibits similar
kinetics, as CFS often have low levels of R-loops²¹. Thus, it remains to be explored whether
transient DNA:RNA hybrids associated with transcription are present at TRC sites and if they
correlate with fork slowing and DNA breaks."

Point 10: “Notably, termination zones exhibited a higher degree of dynamism compared to
initiation zones.”

What is a higher degree of dynamism? And both initiation and termination zones are just
genomic intervals, do author mean that termination zones, as expected, would be more cell-
line and condition dependent? if so, stating it more clearly would be helpful.

Response: we have rewritten the sentence on page 5, lines 17-19:

“We determined the fork direction by connecting the initiation zone to the nearest replication
termination points assisted by a convolutional neural network (Fig. S1B - D and Methods).”

Point 11: “We proposed that fork stalling at inward-moving forks yields centromeric DSB
ends at right-moving forks and telomeric DNA ends at left-moving forks (Fig. 2A).”

Until this, DSB direction were always described as either (from) centromere or (from)
telomere, but now it is mixed with "right" forks, making this sentence difficult to understand.

Response: we uniformed the terminology as “rightward” or “leftward” when describing DNA
replication and transcription directions. The DSB directions were uniform as centromeric or
telomeric. In the result section, we elaborate on the RDC definition: RDCs are now defined
by the direction of replication forks traversing the underlying genomic regions.

Point 12: In terms of purely stylistic remarks, "fork directions" cannot "extend to 1.5Mb",
DSBs are also not a subject of rearrangements, genome is.

Response: We thank the reviewer for pointing out the language problem. We have corrected
these errors accordingly.

Reviewer #3 (Remarks to the Author):

General comment:

Point 1: Using LAM-HTGTS, a technique mapping simultaneously DSB and their orientation,
coupled with a CRISPR-Cas9 inducible system in mouse neural progenitor cells submitted to
replication stress, Ionasz et al. propose that the orientation of DNA replication directs the
orientation of DSB end in respect to centromeres and telomeres. In a second part of the
study, the authors investigate the contribution of transcription and transcription-replication
conflicts (TRC) in the occurrence of DSB at specific loci prone to generate clusters of DNA
breaks (termed RDC). Both aspects are of general interests to understand how DNA damage
and replication stress alter genome stability.

Response:

We thank reviewer 3's positive comments.

Point 2: Although the first part of the study provides a well documented description of DSB
density and orientations according to the directionality of DNA replication, the data
presented in the manuscript regarding the impact of transcription and TRC on the occurrence
of DSB are, in my opinion, much less substantiated (see major issues). Indeed, in most of
the cases/loci studied by the authors, no strong effect of transcription on RDC occurrence is
seen. The authors - and I agree with them - even point out in the discussion that there must
be another key factor determining RDC occurrence that is not transcription per se. Hence, I
found that conclusions drawn in the second part of the manuscript are not well supported by
data and are limited to a few loci that are not behaving similarly. In my opinion, the authors
either need to perform a series of experiment to identify what is contributing to RDC
occurrence - but it might be out of reach in the context of revision - or alternatively, they
could refocus the manuscript on the first part of the paper linking DNA replication
directionality with DSB ends orientation.

Response:

We have removed the second part accordingly, as they became less relevant to the paper,
and strengthened the first part.

Major issues:

Point 3: - Figure 4 and Pages 9-10, lines 15-28 + 1-3. The authors conclude that Head on -
TRC increases DSB density, in a dose-dependent manner, under APH treatment by
analyzing 85 RDC. In the same graph, they also show that Codirectional - TRC decreases
DSB density under APH treatment, in a manner that is not proportional to the dose of APH.

Response:

We realized that the data shown in the original Figure 4 was drawn using the
INTRACHROMOSOMAL DSB at the breakpoint chromosome, which presented a bias towards
the bait DSB direction. To avoid misleading the readers, we presented new figures using only
the interchromosomal DSBs that are not present on the viewpoint chromosome. We replaced
the original Figure 4A with new figures in the revised manuscript (Figure 4A, C, E). In
addition, the exact DNA break density for these regions was described as "DSBs per ten
thousand interchromosomal translocations" in the result section, under page 8, between lines
20 – 26. In addition, we also excluded RDCs, which we cannot directly access the direction

of DNA replication and transcription. The new analyses are presented in Figure 4 without
changing the previous conclusion.

Point 4: Although supporting data look convincing, the authors needs to formulate hypothesis
and to explain these results. For example: Does the APH treatment slowdowns DNA
replication in a way that the replisome never reaches the transcription site? If this is the case,
why would it only be the case for the CD – TRC ? The authors should provide data and/or
further analyses to elucidate that point.

**Response:**

Based on our new DRIPc-seq experiments, we think the presence of dual-strand DNA:RNA
hybrids partially contributed to the bias in the RDC. The hybrids may form behind the
replication forks at the co-directional TRC while being created ahead of the head-on TRC.

In addition, the high-resolution sequencing results indicated that, even at the late CTR, most
genomic regions completed DNA synthesis at the end of S phase. We stated a few
exceptions in the discussion, on page 14, between line 8-14, quote:

“In addition, the high-resolution Repli-seq data indicated that DNA replication is completed at
most CTR regions (Figs. 2, 3, S3, and S4) with a few exceptions at the genomics sequences
underlying *Magi1*, *Ccser1*, and *Grid2* RDCs, where a gap in the CTR was observed (Fig. S3
and S4). This gap is likely due to underreplication at the center of specific RDC-containing
genomes. Hence, a subset of broad late CTR-containing RDC may share the DSB-initiation
mechanism as CFS⁴⁰ (Fig. 7). “

Point 5: - The authors conduct a series of experiments and analyses to identify the
determinants of DSB occurrence in RDC regions, specifically in respect to transcription and
TRC. One of the major issue of the manuscript is that most of the time the authors try to
deduce general features from the analysis of very few (if not a single!) loci. For instance, in
Figure 3, the authors analyzed a single representative gene for early replicating and mid-
replicating regions and draw general conclusion on the differences between early and
mid/late replicating regions.

**Response:**

We understood that the reviewer would like to see more data. The revised manuscript shows
the DNA break density plot for most RDCs in Figures 2, 3, S3, and S4. Due to space
limitations, we omitted three “undefined” RDCs from the plots. We also corrected the
terminology “early-replicating” vs.. “late-replicating” based on the replication feature
determined using the new Repli-seq datasets. We have explained this in the overarching
response at the beginning of this letter and the response to Reviewers 1 and 2.

Point 6: This bias become even more problematic later on, starting from Figure 5, where the
authors argue that transcription is required for RDC occurrence in *Ctnna2* - which seems to
be genuinely the case - but in the same figure they also show that it is not the case for
another locus (*Foxp1*).

**Response:**

We addressed this issue in the response to Reviewer 1, point 6. In the data presented in the
original manuscript, the promoter and enhancer of *Foxp1* (Fig. 5D of the original manuscript)
were not removed. This locus was shown as a control that RDC can be induced in the

Ctnna2-ape neural progenitor cell lines. In addition, all of Figure 5 from the original
manuscript is no longer present in the revised manuscript.

Point 7: Although I agree that at this stage in the manuscript they specifically conclude on the
effect of transcription activity in RDC formation at the Ctnna2, later on in Figure 7, they show
that DSB occurrence at the same locus is not proportional to transcription level.

**Response:**

As the reviewer suggested, we focused on the directionality of transcription and DNA
replication. We removed the original figures 5, 6, and 7 from the revised manuscript to avoid
confusion.

Point 8: Then, in Figure 6, the authors now conclude that it is the full-length transcription that
is required for RDC occurrence in another locus (Nrnx1). Why are these 3 loci behaving
differently? Can we assert a general rule for RDC occurrence? Unfortunately, with the data
presented, I don't think so.

**Response:**

Based on the replication timing, DNA break density, orientation, and the DSB alignment to R-
loops, we concluded that RDCs result from the conflict between linear encountering of DNA
replication and transcription. In contrast to a simple and common cause, the acting
mechanism creating DNA breaks varied. In the case of the RDC at the TTR slopes, these
DNA breaks are presumably generated when reprogramming long-traveling forks. This
process requires DNA nucleases to function at the S phase. DNA breaks are no longer
generated at the long-traveling forks at late CTR. At the early replicating, broad initiation
zones, DNA breaks follow R-loops' density. We believe it is essential to demonstrate that
multiple mechanisms can induce genome fragility. Lacking a common cause of RDC should
not be seen as a weakness.

Many RDCs are specific to neural progenitor cells. We hypothesize that, in NPCs, TTRs
present in genomic regions enriched genes that regulate neuronal functions. To explore this
hypothesis, we conducted a gene ontology enrichment analysis focusing on genes
containing TTRs. Despite the lack of consideration for transcriptional activity in this analysis,

we observed a significant enrichment of genes involved in smell perception encoded within
the genomic region containing TTR (Fig. R8). Given that TTRs are dictated by the location of
initiation zones, and these zones vary with cell type, we postulate that the positioning of
TTRs is cell type-dependent. RDCs may represent the amalgamation of transcription at
TTRs. However, validating this proposed mechanism necessitates experiments that extend
beyond the scope of the current manuscript.

Point 9: In the last figure, the authors demonstrate that activating transcription at an ectopic
locus (Ptn) does not increase RDC whereas it does at another one (Ctnna2) with similar
control of transcription activity. Again, with this result, the authors data strongly suggest that
RDC occurrence cannot be explained in a general manner by transcription, nor it is by the
presence of R-loops (as mentioned in page 10 and Supp Table S1). I have the impression
that the authors describe a series of single-locus events/properties but are unable, at this
stage, to draw or to identify general features for RDC formation.

**Response:**

We understood that reviewer 3 was unsatisfied with the data quantity presented in the
original manuscript. The revised manuscript described the four R-loop rich RDCs, showing
the R-loop position and DNA break density. We also conducted strand-specific DRIPc-seq to
analyze co-transcriptional DNA:RNA hybrids. These examples include multiple genomic loci
(Figure 5 in the revised manuscript). In addition, we provided the number of significant DRIP-
seq peaks and co-transcriptional DNA:RNA hybrids count in the revised Table S4. Lastly,
normalized bigwig files that denote DRIP-seq and DRIPc-seq values are deposited under the
GEO sessions indicated in the manuscript. These data are accessible to readers who need a
complete picture of all RDCs.

Point 10: Finally, I also underscore that the title of their manuscript is: “Transcription-
Replication Conflicts shapes DNA break dynamics”. If the authors want to demonstrate a
direct link between transcription and DNA breaks occurrence, they should provide evidence
that this is a generally common feature of RDC, which is not the case at the moment.

**Response:**

Our manuscript describes the linear interaction between transcription and DNA replication,
not to determine the cause of RDC. To avoid confusion, we revised the title to “Linear
Interaction Between Replication and Transcription Shapes DNA Break Dynamics at
Recurrent DNA Break Clusters” to clarify the focus is on the RDC-containing genomic
regions.

Point 11: - Throughout the manuscript, I find that some informations are hard to find. For
example, in Fig 4C, we know that 85 RDC were taken into account, but how many fell in the
HO vs CD clusters ? Are they similar in size ? Another example is the absence of scale when
authors show DSB density in Figures 2 and 3, which prevents a reader to compare the
various loci in terms of DSB occurrence.

**Response:**

We have specified the exact RDCs categories (“inward-moving”, “unidirectional”, “outward-
moving”, “complex”, and “undefined”) analyzed in each figure. The number of RDC in each
category is indicated in the revised Figure 3E. We included Y-axis scales for all omics figures
presented. The definition of each Y is now explained in the corresponding figure legends.
The average size for co-directional TRC is 331 kb, while head-on TRC is 351 kb. We noted
this number in Figure S6 legends.

Minor points :

Point 12: 1. Introduction, page 3, line 20. Authors are comparing Recurrent DNA break
clusters and Common Fragile Sites and state that they differ in terms of DNA replication. It
would be informative to recapitulate in one sentence the characteristics of CFS to fully
appreciate in which aspects they differ from RDC.

Response: we concluded that only RDCs that display broad and late CTR are similar to CFS,
not the RDCs on the TTR slopes. The relevant texts are in the discussion, on pages 13-14,
and summarized in Figure 7.

Point 13: 2. Results, page 6, lines 5-11. I'm not sure whether the authors also included DSBs
emanating from regions that are different of the CRISPR-Cas9 site in their dataset. Could the
authors explain the rationale and state clearly if they include or not other chromosomal
regions than the CRISPR-Cas9 cleavage site.

Response:

All analyses in the revised manuscript are derived from non-bait viewpoint chromosomes.
The reason that we exclude the DSBs emanating from regions that are different from the
CRISPR-Cas9 site was described in the original manuscript, under the Method section
between page 21, line 23 to page 22, line 5:

“... Only DSB detected at the non-viewpoint chromosome are subjected to statistical
analyses and plotting in Figures 2, 3, 4, 5, 6, and S3, 4, 5, 6. We excluded bait viewpoint-
chromosome for analyses as the Dcen and Dtel recovery rate is unbalanced. The bait
preferentially recovers 15-25% more downstream DSBs at the break site chromosome than
the upstream. Using bait viewpoint chromosome DSB resulted in an overrepresentation of
the centromeric DSB end when the bait had a centromeric orientation. The bait with a
telomeric direction resulted in an overrepresentation of the telomeric DSB end. The bias due
to bait DSB end orientation on the bait viewpoint chromosome was as significant as 20%.
DSB end recovery bias was not present on the non-viewpoint chromosome.”

Point 14: 3. Figure 2 B,D, F and H. DSB density in these figures seems to be associated with
some sort of peak calling or a thresholding methodology as we can see a dash line on graphs
yet I did not find a description of these dashlines in the figure legends, nor I found how it was
determined in the text. This is quite an important point since the authors want to claim that
forks directionality determine DSB orientation. It is even more important given the fact that for
example we can clearly see: (1) signals for Dtel in the example of the Large gene below the
dash line (Fig 2D) and (2) a Dcen peak in Sdk1 (Fig 2H) that is marked with a star even if the
peak stays below the dashed line. Additionally, the authors mention that not all the regions
analyzed behave similarly in a given context (e.g unidirectional: 20/35 exhibit a single peak
DSB signal). To facilitate data visualization and interpretation by readers, I advise the authors
to quantify the enrichment for both Dcen and Dtel in there different contexts and show the
results in a graph where it is possible to see individual regions (e.g violin plots or else) in
addition to the already represented data which are graphical and seems to use only the
thresholding effect. I also found intriguing the absence of scale on the DSB density charts.
This is of paramount importance to allow readers to compare the frequency of DSB in the
different loci shown.

Response:

We appreciate the suggestion from reviewer 3. The DSB distribution at and around the RDC
area is presented (Revised Figures 2, 3, S3, and S4). We also included the annotation of
significant Dcen and Dtel islands in the multiomics plots in Figures 2, 3, S3, and S4. The
RDC calling was conducted by MACS2; the threshold and parameters were described on
Page 22, lines 6-18.

Point 15: 4. Fig2. The frequency of DSB following the expectation is indicated for unidirectional
(20/35) and biphasic replication (5/9) but not for inward and outward moving forks. Does it
means that all regions analyzed in both contexts behaved similarly ?

**Response:**

We analyzed “inward-moving”, “unidirectional”, and “complex” RDC separately for their DNA
break density when treated with aphidicolin concentration (revised Figure 4), and the results
support our prior conclusion. There are only six “outward-moving” RDCs, and the overall
uncton density was very low in them from the aphidicolin dosage experiments (page 9, lines
5-12); hence we cannot conclude their DNA break density. As described before, we cannot
analyze the “undefined” RDC as we cannot access the DNA replication directions.

Point 16: 5. Figure 3, here the authors show DSB density in the presence of aphidicolin +/-
ATR inhibition. It would have been interesting to show, on the same figure at the same scale,
the DSB density in cells without treatment with aphidicolin in order to estimate if there is
already an increase with Aphidicolin treatment alone. This is even more relevant since the
authors quantify DSB amount with various dose of aphidicolin in a subsequent panel (Fig
3E).

**Response:**

We removed the ATR experiments from the revised manuscript with reasons explained in the
overarching response.

Point 17: 6. Figure 3 and Results page 8. The authors conclude in a very general way that
they “demonstrated that genomic underoing early replication in S the phase, as opposed to
regions replicating during the median or late phases, display dormant origin activation upon
ATR inhibition”. This is a bold statement considering that the authors extrapolate this
conclusion from the analysis of a single early replicated locus and a single mid replicated
locus.

**Response:**

We excluded the ATR results; thus, this statement is no longer in the revised manuscript.

Point 18: 7. Figure 3, we can't find informations relative to the duration of treatments (APH
and VE-821) nor if the different doses of aphidicolin employed activate similarly the S phase
checkpoint kinase in NPCs.

**Response:** We removed all ATR-related experiments from the revised manuscript. This point
is no longer applicable for the revised manuscript.

Point 19: 8. Figure 4, it would be informative to depict movement of the transcripton
machinery in the same way than the replication machinery in the various panels.

Response:

The revised figures 2, 3, S3, and S4 showed RefGene direction, not transcription direction detected in neural progenitor cells. We chose to show GROseq data in the revised Figure 4 for readers to access the co-directional vs. head-on analyses described later in the manuscript.

Point 19: 9. Page 9, Results, consider reformulating the sentence at lines 15-16.

Response: we have written the manuscript, and thus, the sentence no longer exists.

Point 20: 10. Page 9, Results, line 20. I don't understand why the authors refer to "The ideal scenario" ?

Response:

We appreciate reviewer 3's comment. It should be "The balanced contribution scenario – when Dcen and Dtel contribute equally". We have corrected the sentence in the revised manuscript on page 12, line 2.

Point 21: 11. Figure 5 is lacking a legend for the colors used in panels C and D.

Point 22: 12. Figure 5, the authors need to explicit, in the results description, if they are referring to samples treated or not with APH (e.g page 11, lines 6-7).

Response: these figures are not present in the revised manuscript.

Point 23: 13. Figure 6: The authors found that the short isoform transcription of *Nrxn1* is not leading to RDC formation, contrary to the long-isoform. Again, we can wonder why. Is it due to the fact DNA replication never reaches the transcription site of the short isoform? What distinguishes the short vs the long isoform in term of transcription?

Response: The new high-resolution Repli-seq data indicated that DNA replication reached the short *Nrxn1* isoform under aphidicolin treatment. The high-resolution Repli-seq data is presented in the revised Figure S4.

Point 24: 14. Figure 7: the comparison between the engineered *Ptn* locus and *Ctnna2* is not clear. Was *Ctnna2* also engineered in the same way and put under the control of Dox responsive element ?

Response: as we described in the previous version, in the Method section, on page 20, line 18, the *Ctnna2* collection was generated by CRISPR/Cas9-mediated deletions. The expression of *Ctnna2* was not controlled under the Dox system. These figures are not present in the revised manuscript.

REVIEWERS' COMMENTS

Reviewer #1 (Remarks to the Author):

The authors made significant changes to their revised manuscript that addresses all of my previous concerns and made new substantive points that help to unify RDC observations notably with the observation that RDCs at TTRs experience dual strand TRC and represent a new class of fragile sites. In this submission, the supplementary table labeling was not clear and will need to be updated. There are also some minor comments below that should help with flow and clarity.

Major points

1. Table references are not accurate or missing a title. For example Table S1 and S6 have no title, there are two Table S2 files: "Table S2. Replication features for APH-treated NPCs" and "Table S2. RDC location, replication pattern of the DNA sequences beneath, and DNA:RNA hybrids peaks within RDC". Not immediately clear which Table reference is missing but table numbering and referencing should be checked again. Table S5 seems to be missing or not labeled properly

Minor points

1. Pg 5 lines 11-17: for consistency between regions and zones it would be more clear to indicate whether untreated and APH-treated NPCs contain the same coverage ranges (or not if that is the case) for the different region/zone descriptors. As written, zones are not impacted by APH but no indication for the timing regions, and lines 5-6 suggest APH advances RT for some genome sites.

2. Pg 5 and 6: the terminology for zones (small) versus regions (large) should be consistently applied throughout the manuscript to aid in clarity. For instance pg 5 line 21 describes timing transition zone which should be in reference to TTR but from figure 2 it looks like zones are discrete areas whereas regions take up much more area, consistent with how RDCs are also spread over a region. Pg. 6 line 21 uses TTR zone which makes this description more confusing. Terminology update may be necessary in the future to better distinguish zones and regions and highlight exceptions (e.g. broad initiation zones).

3. Pg 6 line 17 "...long TTR cannot be given a replication timing..." designation? Need to complete the sentence.

4. Try to keep consistent with which tense to use. Pg 9 line 2 uses past tense but line 5 uses present tense. There are several other instances throughout the manuscript to adjust as well.

5. Pg 11 lines 12-24: this paragraph should reference 4C and 4E for the other two genes.

Reviewer #2 (Remarks to the Author):

The authors made a lot of effort to gather new experimental data and very substantially revised the manuscript to include mine and my fellow reviewers' remarks. I am satisfied with the changes that were made and the explanations provided and I think the manuscript is now fit for the publication in Nature Communications.

Reviewer #3 (Remarks to the Author):

The authors proposed a revised version of their manuscript that is answering to the questions raised during the revision.

Response to reviewers

Manuscript ID: **NCOMMS-23-39112A**

Manuscript title: Linear Interaction Between Replication and Transcription Shapes DNA Break Dynamics at Recurrent DNA Break Clusters

REVIEWERS' COMMENTS

Reviewer #1 (Remarks to the Author):

The authors made significant changes to their revised manuscript that addresses all of my previous concerns and made new substantive points that help to unify RDC observations notably with the observation that RDCs at TTRs experience dual strand TRC and represent a new class of fragile sites. In this submission, the supplementary table labeling was not clear and will need to be updated. There are also some minor comments below that should help with flow and clarity.

Major points

1. Table references are not accurate or missing a title. For example Table S1 and S6 have no title, there are two Table S2 files: "Table S2. Replication features for APH-treated NPCs" and "Table S2. RDC location, replication pattern of the DNA sequences beneath, and DNA:RNA hybrids peaks within RDC". Not immediately clear which Table reference is missing but table numbering and referencing should be checked again. Table S5 seems to be missing or not labeled properly.

Response: we thank reviewer's carefulness. We have included the description page for all tables. We also renamed the tables to "supplementary data" per Nature Communications rule.

Minor points

1. Pg 5 lines 11-17: for consistency between regions and zones it would be more clear to indicate whether untreated and APH-treated NPCs contain the same coverage ranges (or not if that is the case) for the different region/zone descriptors. As written, zones are not impacted by APH but no indication for the timing regions, and lines 5-6 suggest APH advances RT for some genome sites.

2. Pg 5 and 6: the terminology for zones (small) versus regions (large) should be consistently applied throughout the manuscript to aid in clarity. For instance pg 5 line 21 describes timing transition zone which should be in reference to TTR but from figure 2 it looks like zones are discrete areas whereas regions take up much more area, consistent with how RDCs are also spread over a region. Pg. 6 line 21 uses TTR zone which makes this description more confusing. Terminology update may be necessary in the future to better distinguish zones and regions and highlight exceptions (e.g. broad initiation zones).

Response: we have unified the terminology as timing transition "region". Regarding on initiation zones, it was defined by David Gilbert's team. We intend to keep its original name as defined.

3. Pg 6 line 17 "...long TTR cannot be given a replication timing..." designation? Need to complete the sentence.

Response: we meant "long TTR cannot be given a replication timing."

4. Try to keep consistent with which tense to use. Pg 9 line 2 uses past tense but line 5 uses present tense. There are several other instances throughout the manuscript to adjust as well.

Response: we corrected the present tense issue to the best of our ability.

5. Pg 11 lines 12-24: this paragraph should reference 4C and 4E for the other two genes.

Response: we added the references to 4C and 4E.

Reviewer #2 (Remarks to the Author):

The authors made a lot of effort to gather new experimental data and very substantially revised the manuscript to include mine and my fellow reviewers' remarks. I am satisfied with the changes that were made and the explanations provided and I think the manuscript is now fit for the publication in Nature Communications.

Response: we thank Reviewer 2's positive comments. We are glad our revision has addressed all your questions.

Reviewer #3 (Remarks to the Author):

The authors proposed a revised version of their manuscript that is answering to the questions raised during the revision.

Response: we thank Reviewer 3's positive comments. We are glad our revision has addressed all your questions.